# A novel and robust feature selection method with FDR control for omics-wide association analysis

**Zhibo Chen, Zi-Tong Lu, Xue-Ting Song, Yu-Fan Gao, Jian Xiao**[ID]*

School of Statistics and Mathematics, Zhongnan University of Economics and Law, Wuhan, Hubei, People's Republic of China

¶ Membership list can be found in the Acknowledgments section.
* xiaoj771@sina.com

**Data availability statement:** All relevant data sets are in the Supporting information files of the paper.

## Abstract

Omics-wide association analysis is a very important tool for medicine and human health study. However, the modern omics data sets collected often exhibit the high-dimensionality, unknown distribution response, unknown distribution features and unknown complex association relationships between the response and its explanatory features. Reliable association analysis results depend on an accurate modeling for such data sets. Most of the existing association analysis methods rely on the specific model assumptions and lack effective false discovery rate (FDR) control. To address these limitations, the paper firstly applies a single index model for omics data. The model shows robust performance in allowing the relationships between the response variable and linear combination of covariates to be connected by any unknown monotonic link function, and both the random error and the covariates can follow any unknown distribution. Then based on this model, the paper combines rank-based approach and symmetrized data aggregation approach to develop a novel and robust feature selection method for achieving fine-mapping of risk features while controlling the false positive rate of selection. The theoretical results support the proposed method and the analysis results of simulated data show the new method possesses effective and robust performance for all the scenarios. The new method is also used to analyze the two real datasets and identifies some risk features unreported by the existing finds.

## Introduction

Advances in high-throughput omics technologies, such as metagenomics sequencing and DNA methylation or protein microarrays, have revolutionized research in medicine. One major use of such technologies is conducting omics-wide association analysis to identify relevant omics features, such as microbial genes and DNA methylation variants, from a large pool of candidate features by analyzing their association with a phenotype of interest, e.g., patient prognosis or response to medical treatment. The identified features can help understand the disease mechanisms, subject to further validation studies, or be used to form more

**Funding:** This work was supported by the National Natural Science Foundation of China (NSFC) (no. 11801571, no. 12171483 and no. 61773401).

**Competing interests:** The authors have declared that no competing interests exist.

accurate prediction models for personalized medicine [1]. The increasing availability of massive human genomic data sets makes the dimensionality of omics features much larger than its sample size, which also poses new challenges to statistical analysis [2–4]. However, the modern omics data sets collected often not only own the high-dimensionality property but also exhibit unknown distribution response, unknown distribution features and unknown complex associated relationships between the response and its explanatory features. Reliable association analysis results rely on an accurate high-dimensional modeling for such data sets. Such an involved modelling task can be based on high-dimensional single index model (SIM) method. The single index model has been the subject of extensive investigation in both the statistics and biology literatures over the last few decades. It generalizes the linear model to scenarios where the regression function can be any monotonic link function including the nonlinear functions. High-dimensional single index models have also attracted interest with various authors studying variable selection, estimation and inference using penalization schemes [5–14]. However, almost of all these methods do not provide a false discovery rate (FDR) controlled multiple testing procedure for simultaneously testing the significance of model coefficients and can not perform FDR controlled feature selection. In particular, for the feature selection of high-dimensional single index models, Rejchel et al. [32] proposed the cross-validation Ranklasso method, along with its modified versions: the thresholded Ranklasso and the weighted Ranklasso methods. However, the precision of feature selection of all these regularization-based methods depends on the tuning parameter in the penalty function and sample size. Given a specific sample size, the relationship between the false feature selection rate (e.g. FDR) and the tuning parameter value remains unknown. Therefore, it is challenging to use the tuning parameter to control the FDR for the selection results of Ranklasso-type methods. As a result, the Ranklasso-type methods are unable to effectively conduct FDR-controlled feature selection. In addition, the multiple testing work with FDR control of few methods relies on $p$-values with Benjamini and Hochberg (BH) correction [15], and may fail to control FDR in the presence of complex and strong dependence structure among features. Moreover, high-dimensionality of features makes such approach to have lower power due to large-scale adjustment burden. Thence using high-dimensional single index model to develop an effective and robust FDR controlled feature selection method becomes very desired.

As we know, in the existing literature, the FDR controlled feature selection methods for high dimensional models can be achieved mainly via the following three approaches.

1. Knockoff filter-based approach.
   Barber and Candes [16] firstly introduced the knockoff filter, a new feature selection procedure controlling the FDR in the statistical linear model whenever there are at least as many observations as variables. This method achieves exact FDR control in finite sample settings no matter the design or covariates, the number of variables in the model, or the amplitudes of the unknown regression coefficients, and does not require any knowledge of the noise level. Following the knockoff filter framework, Candes et al. [17] proposed a model-X knockoff FDR control method. However, this method requires both complete knowledge of the joint distribution of the design matrix and repeated derivation of the conditional distributions. The development of methods to construct exact or approximate knockoff features for a broader class of distributions is a promising area of active research [18,19].
2. Symmetrized data aggregation (SDA) approach.
   To simultaneously test the significance of the regression coefficients in high-dimensional linear regression model, Du et al. [20] firstly proposed a data splitting-based method "SDA" to select features with FDR control. The key idea of the proposed

method is to apply the sample-splitting strategy to construct a series of statistics with marginal symmetry property and then to utilize the symmetry for obtaining an approximation to the number of false discoveries. The SDA approach consists of three procedures.

 a. The first procedure splits the sample into two parts, both of which are utilized to construct statistics to assess the evidence of the regression coefficient against the null.

 b. The second procedure aggregates the two statistics to form a new ranking statistic fulfilling symmetry about zero properties under the null.

 c. The third procedure chooses a threshold along the ranking by exploiting the symmetry about zero property between positive and negative null statistics to control the FDR.

3. Stability selection approach.
Stability selection [21], an innovative variable selection algorithm, relies on resampling datasets to identify variables. The core principle of this approach involves repeatedly applying the variable selection method to resampling subsets of the data, thereby defining a variable as stable variable if it is frequently selected. It can help a feature selection method like lasso to improve its performance. Another good property of stability selection is that it provides an effective way to control false discovery rate (FDR) in finite sample cases provided that its tuning parameters are set properly. Due to its versatility and flexibility, stability selection has been successfully applied in many domains such as gene expression analysis [22–25].

Additionally, FDR controlled feature selection procedure can also be implemented by many other statistical inference methods (eg, regression-based modeling, two-sample testing, and statistical causal mediation analysis) [26–31]. However, directly applying these statistical methods to analyze the omics data is usually underpowered and sometimes can render inappropriate results (The selection results may contain too many false discoveries).

In this paper, we firstly employ a single index model (SIM) to model omics data. The considered SIM is robust in performance of the two aspects. One aspect involves the relationships between the response variable and a linear combination of covariates connected by an unknown monotonic link function, while the other aspect involves the random error and covariates following unknown distributions. In this paper, our plan is applying the above FDR-controlled feature selection approaches and the rank-based approach [32] for high-dimensional SIM to develop a novel and robust FDR-controlled feature selection method. More specifically, our plan involves constructing a single index model based on the rank-based approach, in which the transformed response samples are dependent. However, such dependency violates the independence assumption of the response samples underlying the Knockoff filter-based approach. Consequently, the Knockoff filter approach is not applicable to our rank-based plan. Regarding the stability selection approach, the existing theory, which significantly informs its implementation, provides relatively weak bounds on FDR, leading to a reduced number of true positives [33–36]. Furthermore, stability selection necessitates users to specify two out of three parameters: the target FDR, a selection threshold, and the expected number of selected features. Numerous studies have demonstrated that stability selection is sensitive to these choices, complicating tuning for optimal performance [34,36–39]. In summary, stability selection not only requires numerous parameter settings but also is excessively conservative, leading to a reduced number of false positives (FP) at the expense of true positives (TP). Hence, the stability selection approach is not an appropriate choice for our plan. The SDA approach is independent of $p$-values, owns higher power performance, and

requires significantly fewer tuning parameters, which allows it to possess several useful theoretical properties. This motivates us to adopt the SDA approach within the context of the rank-based SIM model. Consequently, we leverage both the rank-based approach [32] and the SDA approach to develop a robust feature selection method for SIM, which we subsequently apply to fine-mapping of omics data while controlling the false positive rate of selection. Notably, the proposed method does not depend on $p$-values. Additionally, we utilize theoretical results to validate the effectiveness of the proposed method.

Finally, we design extensive simulation studies to compare the proposed method with the competing methods.The simulation results demonstrate that the proposed method effectively controls FDR across all scenarios. Additionally, the results indicate that when the sample size is moderate and errors stem from a heavy-tailed Cauchy distribution, or when the relationship is non-linear, or when the relationship is non-linear and errors follow a heavy-tailed Cauchy distribution, the proposed method outperforms other competing methods in terms of power performance. For small sample size scenario, compared to the proposed method, other competing methods may either be underpowered or render inappropriate results by having an inflated FDR than the nominal FDR threshold. These results indicate our method owns effective and robust performance for all the scenarios. The proposed method is also applied to the analysis of two real data sets and identifies some casual features unreported by the existing finds.

## Materials and methods

This section firstly reviews rank-based approach single index model (SIM) for omics data and then provides parameter estimation methods for SIM. Finally, a multiple testing procedure with FDR control is given for simultaneously testing the coefficients of high-dimensional single index model.

### Review of based on rank approach single index model

Let $X = (X_{i,j})_{n \times p}$ denote observed $p$ omics features matrix on $n$ samples; $Y = (Y_i)_{n \times 1}$ denote the response vector such as disease status, gene expression and so on. Assume that the means of all the $p$ features are zero, Rejchel et al. [32] focused on the single index model (1) without intercept

$$Y_i = g(\sum_{j=1}^{p} X_{i,j}\beta_j, \varepsilon_i), i = 1, \ldots, n, \tag{1}$$

where $\beta = (\beta_1, \ldots, \beta_p)^\top$ denotes the coefficients associated with the omics features on the response $Y$, $g(\cdot)$ is an unknown monotonic link function. No assumptions are made on the form of the monotonic link function $g$, or the distribution of the error $\varepsilon_i$, or the distribution of all the $p$ features, thence model (1) is robust for modeling the omics data. Their purpose is to perform feature selection to identify the feature set $S = \{j : \beta_j \neq 0, j = 1, \ldots, p\}$ within the framework of model (1). For this purpose, Rejchel et al. [32] utilized a rank-based lasso approach to sparsely estimate $S$ by optimizing the problem

$$\widehat{\theta}_{\text{RLasso}} = \arg\min_{\theta}\left[\frac{1}{2n}||Y^R - X\theta||_2^2 + \lambda||\theta||_1\right], \tag{2}$$

where $Y^R = (Y_i^R)_{n\times 1}$ with $Y_i^R = \frac{R_i}{n} - 0.5$; $R_i = \sum_{j=1}^{n} I(Y_j \leq Y_i)$ denotes the rank values of the response actual values $Y_i$ by their centered ranks; $||\cdot||_1$ and $||\cdot||_2$ denotes $l_1$ and $l_2$ norm, respectively; $\lambda > 0$ is a data-dependent tuning parameter of $l_1$ norm type lasso penalty.

Rejchel et al. [32] highlighted that the rank-based lasso method, as defined in the optimization problem (2), does not estimate true regression coefficients $\beta$ in model (1). However, given some assumptions, they defined a parameter $\theta^0$ by the optimization problem without introducing the penalty

$$\theta^0 = \arg\min_{\theta \in \mathbb{R}^p} E[Q(\theta)], \quad Q(\theta) = \frac{1}{2n} \| Y^R - X\theta \|_2^2,$$

which is related to the true vector of regression coefficients $\beta$. It is revealed that, under certain standard assumptions, the support $S_{\theta^0} = \{j : \theta_j^0 \neq 0, j = 1, \ldots, p\}$ of $\theta^0$ coincides with the support $S = \{j : \beta_j \neq 0, j = 1, \ldots, p\}$ of $\beta$. Furthermore, Rejchel et al. [32] demonstrated that the estimator $\widehat{\theta}_{\text{RLasso}}$ is a consistent estimate of $\theta^0$, and thus can be utilized to identify $S$. More detailed information can be found in the work [32].

## Omics-wide association analysis based on robust single index model

In real omics data analysis, the response variable $Y$ often follows an unknown distribution, which may result in non-linear associated relationships between the response variable $Y$ and the linear combination of omics features $X$. In order to simultaneously account for the unknown non-linear associated relationships, unknown distribution of response and unknown distribution of features, we employ model (1) to robustly model the omics data.

Based on high-dimensional model (1) with a large number of features ($p \geq n$), we focus on the multiple testing problem (3) under the null hypothesis:

$$H_i^0 : \beta_i = 0, \ i = 1, \ldots, p, \tag{3}$$

to identify the features associated with the response $Y$ while controlling for false positives. Because the above section has shown the support $S_{\theta^0} = \{j : \theta_j^0 \neq 0, j = 1, \ldots, p\}$ of $\theta^0$ coincides with the support $S = \{j : \beta_j \neq 0, j = 1, \ldots, p\}$ of $\beta$ in model (1), the multiple testing problem (3) under the null hypothesis becomes

$$H_i^0 : \theta_i^0 = 0, \ i = 1, \ldots, p. \tag{4}$$

The objective of this paper is developing a FDR controlled multiple testing procedure for statistical inference problem (4) under high-dimensional model (1) framework to perform omics-wide association analysis.

## Estimation methods for the parameter $\theta^0$

For the statistical inference problem (4), this section introduces estimation methods for the true parameter $\theta^0$ from the high-dimensional model (1) with a large number of features ($p > n$) and the low-dimensional model (1) with a small number of features ($p < n$), respectively. For high-dimensional scenario, we employ the rank-based lasso method in (2) to obtain the sparse parameter estimate $\widehat{\theta}_{\text{RLasso}}$. Given the observed variables $X$ and the transformed variable $Y^R$, the optimization problem (2) can be implemented using the R package *glmnet*.

For low-dimensional scenario, we use rank-based ordinary least squares method to estimate the parameters $\theta^0$ by solving the optimization problem defined in equation (5)

$$\widehat{\theta}_{\mathrm{ROLS}} = \arg\min_{\theta}\Big[\frac{1}{2n}\|Y^R - X\theta\|_2^2\Big],\tag{5}$$

and obtain the parameter estimators

$$\widehat{\theta}_{\mathrm{ROLS}} = (X^{\top}X)^{-1}X^{\top}Y^R.\tag{6}$$

If there exists strong correlations among $X$, causing the ROLS method to suffer from multi-collinearity and become ineffective, we first use the variation inflation factor (VIF) to assess the presence of multicollinearity, and then based on the VIF value, we eliminate the problematic features. Subsequently, we apply the ROLS method to estimate the parameters of the model constructed by the remaining features. Especially, if there exists a continuous type confounder causing multicollinearity, we can consider stratified approach to solve this problem. More specifically, we can firstly categorize this continuous confounder variable into multiple, say four, categories, or into discrete type variable. Then we can fit four conditional rank-based regression models across the four categories of confounders. Furthermore, take the weighted average of the four parameter estimates from the four conditional models, as the final parameter estimates. We believe such approach can remove most of the bias due to confounding. The properties of the estimators $\widehat{\theta}_{\mathrm{ROLS}}$ and $\widehat{\theta}_{\mathrm{RLasso}}$ can be found in the "Theoretical properties" section (Given in the following).

*Remark 1.* In the following section, we use the SDA approach to develop a FDR controlled feature selection method. The SDA approach requires under null hypothesis, the distribution of the estimator (obtained under low-dimensional scenario) should be symmetric around zero or asymptotically symmetric around zero. For the low-dimensional case, when the number of features is close to or not sufficiently small compared to the sample size, the thresholded Ranklasso method [32] should be used to obtain an estimator with asymptotically symmetric properties around zero under null hypothesis. (Note that under null hypothesis, the distributions of the regular lasso and the weighted Rank-lasso estimators [32] are usually not asymptotically symmetric around zero). However, in our simulation studies, we observed that the number of features is relatively small across all scenarios. Additionally, when the thresholded Ranklasso method is utilized in low-dimensional cases, its power is significantly lower compared to using the ROLS method. This is because the thresholded Ranklasso method conducts feature selection again in low-dimensional scenarios, resulting in a decrease in the number of true signals. Thence we adopt ROLS method for parameter estimation under low-dimensional scenario.

## FDR controlled robust feature selection procedure

In this section, we apply the SDA approach [20] illustrated in the "introduction" section to problem (4) for testing the parameter $\theta^0$, and develop a corresponding FDR controlled feature selection procedure. For the two-part independent samples split in the first procedure of SDA, we employ the RLasso and ROLS methods to estimate the parameters, respectively. By the first part sample, we utilize rank-based lasso to sparsely estimate the parameters $\theta^0$ to identify fewer candidate omics features associated with the response. The purpose of reducing feature dimensionality is to alleviate the burden of multiple testing for the third procedure of SDA. Subsequently, for the second part sample, we employ the rank-based ordinary least square method to obtain a more precise estimate under the low-dimensional case. Finally, we

utilize the estimates obtained by the two-part samples to construct the FDR-controlled feature selection procedure.

More specifically, the proposed procedure is outlined as follows.

- Step 1: Splitting samples.
  Given the ratio $\gamma$, the sample is randomly split into two independent disjoint parts $\zeta_1$ and $\zeta_2$ with sample sizes $n_1$ and $n_2$, respectively, where $n_1 + n_2 = n$ and $n_1/n = \gamma$. The work [20] utilized the simulation studies to verify that the setting $\gamma = 2/3$ is often the most powerful for SDA approach, thus we also use this ratio to split data for simulation studies and real data analysis.
- Step 2: Selecting the candidate omics feature set using the first part sample $\zeta_1$.
  The RLasso method is employed to obtain the estimates $\widehat{\theta}_{\mathrm{RLasso}} = \{\widehat{\theta}_{\mathrm{RLasso},1}, \ldots, \widehat{\theta}_{\mathrm{RLasso},p}\}$ of the parameter $\theta^0$ of $p$ features by the first part sample $\zeta_1$. These non-zero estimates are further used to get the candidate feature set $S_0 = \{i : \widehat{\theta}_{\mathrm{RLasso},i} \neq 0, i = 1, \ldots, p\}$. Based on the first part sample $\zeta_1$, we use the candidate features $X_{\zeta_1}^{S_0}$ to construct a low-dimensional single index model

$$Y_{\zeta_1} = g(X_{\zeta_1}^{S_0}\beta, \ \varepsilon). \tag{7}$$

  Then the ROLS method is employed for model (7). to obtain the estimates $\widehat{\theta}_{\mathrm{ROLS},i}^1$ of $\theta_i, i \in S_0$.
- Step 3: Similar to the 2th step, firstly use the candidate features $X_{\zeta_2}^{S_0}$, based on the second part sample $\zeta_2$, to construct a low-dimensional single index model, then utilize this model to obtain the estimates $\widehat{\theta}_{\mathrm{ROLS},i}^2$ of $\theta_i, i \in S_0$.
- Step 4: Constructing the test statistics under null for problem (4).
  Under null, we utilize the estimates from the two different part samples to construct the statistics $T_{1,i} = \frac{\widehat{\theta}_{\mathrm{ROLS},i}^1}{\varpi_i^1}$ and $T_{2,i} = \frac{\widehat{\theta}_{\mathrm{ROLS},i}^2}{\varpi_i^2}$, with $i \in S_0$ and $T_{2,i} = 0$ for $i \notin S_0$. Here, $\varpi_i^j$ is considered as $\widehat{\mathrm{Se}}(\widehat{\theta}_{\mathrm{ROLS},i}^j)$ and is viewed as a scaling constant that is independent of $\widehat{\theta}_{\mathrm{ROLS},i}^1$ and $\widehat{\theta}_{\mathrm{ROLS},i}^2$. From the Theorem 1 (Given in the following Theory section), the variance of $\widehat{\theta}_{\mathrm{ROLS},i}^k$ is very complicated so that it is difficult for us to compute it. It is well known that the bootstrap re-sampling method can help consistently estimate the variance of the unknown distribution or the complex distribution variable [40–42]. Thence we employ the bootstrap re-sampling method to obtain $\widehat{\mathrm{Se}}(\widehat{\theta}_{\mathrm{ROLS},i}^k)$, where $k = 1, 2$.
- Step 5: Aggregating the test statistics.
  Aggregate the two statistics obtained in the 4th step to form a new ranking statistic that satisfies the property of being symmetric around zero under the null hypothesis:

$$W_j = T_{1,j}T_{2,j}, j = 1, \ldots, p.$$

  *Remark 2:* Because $\widehat{\theta}_{\mathrm{ROLS},i}^k$ without being thresholded follows asymptotically normal distribution with the mean being $\theta_i^0$, $T_{k,i}, k = 1, 2$ has asymptotically normal distribution with the mean being 0 under null $H_i^0 : \theta_i^0 = 0$. This demonstrates that $W_j = T_{1,j}T_{2,j}$ possesses the property of being symmetric around zero. Intuitively, the positive and larger $W_j$ values indicate strong evidence against the null hypothesis, while the negative $W_j$ values most likely correspond to null cases.
- Step 6: Choosing the threshold.
  Given the nominal level $\alpha \in [0, 1]$, a threshold is chosen to exploit the symmetrical property between positive and negative null statistics in order to control the False Discovery

Rate (FDR) at the level $\alpha$:

$$L = \inf\left\{t > 0 : \frac{\sharp\{j : W_j \leq -t\}}{\sharp\{j : W_j \geq t\} \vee 1} \leq \alpha\right\},$$

where $\sharp$ denotes the number of elements in a set. The rejected features are given by $\{j : W_j \geq L, 1 \leq j \leq p\}$.

- Step 7: Robustly selecting the discovery set.

  The goal of this step is to further stabilize the selection result. The reason is that the selection result is not stable and can vary substantially across different data splits due to the inflation of variances of the estimated coefficients by randomly splitting the data into two halves. Therefore, we propose the following procedure to robustly select features.

  Suppose we repeat the above 6-step data-splitting procedure $B$ times independently. Each time, the set of selected features is denoted as $S_b, b \in \{1, 2, \ldots, B\}$. For the $j$-th feature, we define the empirical inclusion rate $\widehat{I}_j$ as:

  $$\widehat{I}_j = \frac{1}{B}\Sigma_{b=1}^{B}I(j \in S_b), j = 1 \ldots, p.$$

  Sort the features based on their empirical inclusion rates in increasing order. Denote the sorted empirical inclusion rates as $0 \leq \widehat{I}_{(1)} \leq \widehat{I}_{(2)} \leq \cdots \widehat{I}_{(p)}$. Then select the features $\widehat{S} = \{j : \widehat{I}_j \geq \widehat{I}_{(p - \tau_{\mathrm{median}} + 1)}\}$, where $\tau_{\mathrm{median}}$ denotes the median size of the selected features over the $B$ runs. In general, a larger B value will result in more stable feature selection. Through simulation studies, the performance of the proposed procedure using $B = 15$ is very similar to that using larger $B$ values ($B > 15$). Therefore, in order to save running time, we set $B = 15$ for both the simulation and real data analysis.

To facilitate comprehension of the aforementioned procedure, a simple flow path is provided below.

1. Randomly splitting the entire samples into the two independent parts.
2. By the first part sample, we employ Ranklasso to get the parameter estimates $\widehat{\theta}_{\mathrm{Rlasso}}$ and use $\widehat{\theta}_{\mathrm{Rlasso}}$ to identify fewer candidate omics features associated with the response. Denote the selected candidate feature set as $\widehat{S}$. The goal of reducing the dimensionality of features is to achieve more accurate parameter estimation in a low-dimensional setting. The well-performing estimators are then utilized to construct the test statistics.
3. Based on the first part sample, we apply ROLS method for low-dimensional single index model (built on fewer candidate omics features set) to obtain a more precision estimates $\widehat{\theta}_{\mathrm{ROLS},j}^{1}, j \in \widehat{S}$ and let $\widehat{\theta}_{\mathrm{ROLS},j}^{1} = 0, j \notin \widehat{S}$.
4. Based on the second part sample, we apply ROLS method for low-dimensional single index model (built on fewer candidate omics features set) to obtain a more precision estimates $\widehat{\theta}_{\mathrm{ROLS},j}^{2}, j \in \widehat{S}$ and let $\widehat{\theta}_{\mathrm{ROLS},j}^{2} = 0, j \notin \widehat{S}$.
5. The estimators $\widehat{\theta}_{\mathrm{ROLS}}^{1}$ and $\widehat{\theta}_{\mathrm{ROLS}}^{2}$ (gotten by the two part indenpendent samples) are utilized to construct statistics to assess the evidence of the regression coefficient against the null, respectively.
6. Aggregating the two statistics to form a new ranking statistic fulfilling symmetry about zero properties under the null.
7. Choosing a threshold along the ranking by exploiting the symmetry about zero property between positive and negative null statistics to control the FDR.
8. Stabilizing the selection results by multi-splitting procedure.

*Remark 3*. The proposed procedure is denoted as SIM-FDR. From the above procedure, FDR control of SIM-FDR method does not rely on *p*-values and requires that the distribution of the aggregated statistics $W_j$ should exhibit symmetry about zero under the null hypothesis. In fact, SIM-FDR does not require that the distribution of $W_j$ must be normal or asymptotically normal. It only requires that the distribution should exhibit the property of symmetry about zero. Thus, SIM-FDR may not heavily rely on the sample size. The simulation results in small sample settings verify this point, indicating that SIM-FDR is more robust than its competitors for a wide range of scenarios, as achieving asymptotic symmetry is much easier in practice compared to achieving asymptotic normality.

## Theoretical properties

This section discusses the finite-sample and asymptotic FDR control properties of the proposed SIM-FDR. Given the intuitive nature of the 7th step of SIM-FDR, our focus is solely on proving the FDR control properties of the previous 6-step SDA approach. Prior to presenting all the theoretical results, we firstly provide the assumptions and definitions. The proofs of all the theorems are provided in S1 File (Supporting information section). Recall the definition of the true parameter $\theta^0$ as given in the "Review" section above.

$$\theta^0 = \arg\min_{\theta \in \mathbb{R}^p} E[Q(\theta)], \quad Q(\theta) = \frac{1}{2n}\|Y^R - X\theta\|_2^2.$$

### Required basic assumptions

**Assumption 1.** Assume that $(Y_i, X_{i,\cdot}, \varepsilon_i), i = 1, \ldots, n$ are independent and identically distributed with $X_{i,\cdot}$ denoting the predictor vector $X_{i,\cdot} = (X_{i,1}, \ldots, X_{i,p})$ of *i*-th sample, the distribution of $X_i$ is absolutely continuous with $X_i$ denoting the *i*-th predictor variable, $\mathbb{E}(X_i) = 0$, and the noise variable $\varepsilon_i$ is independent of $X_{i,\cdot}$.

**Assumption 2.** We assume that for each $\theta \in \mathbb{R}^p$, the conditional expectation $\mathbb{E}(X_{i,\cdot}\theta|X_{i,\cdot}\beta)$ exists and $\mathbb{E}(X_{i,\cdot}\theta|X_{i,\cdot}\beta) = d_\theta X_{i,\cdot}\beta$ for a real number $d_\theta \in \mathbb{R}$.

**Assumption 3.** We assume the cumulative distribution function $F$ of the response variable $Y_i$ is increasing and $g$ in model (1) is increasing with respect to the first argument $X_{i,\cdot}\beta$.

**Assumption 4.** Let $p_0 = |S|$ denote the number of elements in support set *S*. We suppose that the significant predictors $X_{i,\cdot}^S$ is sub-gaussian with the coefficient $\tau_0 > 0$, i.e. for each $u \in \mathbb{R}^{p_0}$ we have $\exp(X_{i,\cdot}^S u) \le \exp(\tau_0^2 u^\top u/2)$, where $X_{i,\cdot}^S$ denotes the sub-matrix of $X_{i,\cdot}$ on the column indices from *S*. Moreover, the irrelevant predictors are univariate sub-gaussian, i.e. for each $a \in \mathbb{R}$ and $j \notin S$, we have $\exp(aX_{i,j}) \le \exp(\tau_j^2 a^2/2)$ for positive numbers $\tau_j$. Finally, we denote $\tau = \max(\tau_0, \tau_j, j \notin S)$.

*Remark 4*. No other assumptions are made on the distribution of the noise variable $\varepsilon$. Assumption 2 is a standard condition in the literature on the single index model and can be found in the work [32]. The Assumption 3 satisfies the needs of rank-based approach. The regular sub-gaussian condition is added on the feature matrix *X* by the Assumption 4.

Consider the model (1), Rejchel et al. [32] show under Assumptions 1 and 2, the conclusion

$$\theta^0 = \gamma_\beta \beta$$

holds with $\gamma_\beta = \arg\min_{d \in \mathbb{R}} \mathbb{E}[Q(d\beta)]$ being a positive constant; under assumption 3, the signs of $\beta$ coincide with the signs of $\theta^0$ and

$$S = \{j : \beta_j \ne 0\} = \{j : \theta_j^0 \ne 0\}.$$

This result indicates the support of $\beta$ in model (1) coincides with that of the rank approach-based true coefficient $\theta^0$. Such conclusions tell us we can perform feature selection for model (1) by using the estimates of $\theta^0$.

## Definitions of the cone invertibility factor (CIF)

In our article, the validation of SIM-FDR relies on the consistency of parameter estimation for the Rank-lasso problem (2), which in turn supports the feature selection screening property as discussed in problem (2). In the high-dimensional setting, Rejchel et al. [32] have demonstrated that ensuring the consistency of parameter estimators generated by the Rank-lasso penalty requires satisfying the cone invertibility factor (CIF) condition defined on the feature matrix $X$.

Let $\theta^S$ and $\theta^{\bar{S}}$ be the restrictions of the vector $\theta \in \mathbb{R}^p$ to the indices from $S$ and $\bar{S}$ with $\bar{S} = \{1, \ldots, p\} \backslash S$, respectively. Now, for $\xi > 1$ we consider a cone

$$\mathcal{C}(\xi) = \{\theta \in \mathbb{R}^p : ||\theta^{\bar{S}}||_1 \leq \xi ||\theta^S||_1\}.$$

Define the population version of CIF to be

$$F_q(\xi, X) = \inf_{0 \neq \theta \in \mathcal{C}(\xi)} \frac{p_0^{1/q} ||\mathbb{E}(X_{i,}^\top X_{i,})\theta||_\infty}{||\theta||_q}, \tag{8}$$

for a sharp formulation of convergency results for all $l_q$ norms with $q \geq 1$.

## FDR control

Before presenting the theorems for controlling finite-sample FDR and asymptotic FDR of the proposed procedure, we first establish the property of asymptotic symmetry around zero for the test statistics $W_j$ and the feature screening property for the candidate feature selection result obtained by the RLasso method in the second step of the proposed procedure. These results are essential for demonstrating FDR control.

**Asymptotically symmetry around zero property of the statistics $W$.** We prove the conclusion that the statistics $W_j$ is asymptotically symmetric around 0 under null hypothesis. Obviously, the distribution of the statistics $W_j$ depends on that of the estimator $\widehat{\theta}_{\text{ROLS}}$ gotten at low-dimensional model (1) scenario.

**Theorem 1.** Suppose that Assumptions 1, 2, 3 and 4 are satisfied, $\mathbb{E}|X_i|^4 < \infty, i = 1, \ldots, p$ with $X_i$ denoting the $i$-th predictor variable and the covariance matrix $\Sigma_X$ of the features $X$ is positive definite, we have the following conclusions for the OLS estimator $\widehat{\theta}_{\text{ROLS}}$ of $\theta^0$ under low-dimensional model scenario,

$$\frac{\widehat{\theta}_{\text{ROLS},j} - \frac{n}{n-1}\theta_j^0}{\sqrt{\text{Var}(\widehat{\theta}_{\text{ROLS}j})}} \to_d N(0,1), \quad j = 1, \ldots, p,$$

where $\text{Var}(\widehat{\theta}_{\text{ROLS}j}) = [\Sigma_X^{-1} D \Sigma_X^{-1}]_{j,j}$ with $[\Sigma_X^{-1} D \Sigma_X^{-1}]_{j,j}$ denoting the $j$-th diagonal element in the matrix $[\Sigma_X^{-1} D \Sigma_X^{-1}]$, and $D$ is defined by the Lemma 3 given in the supplementary materials (Given in Supporting information). Furthermore, under null hypothesis $\theta_j^0 = 0$, the statistics $W_j$ has the asymptotically symmetry around zero property.

**Sure screening property for the candidate feature selection result.** In this section, we prove the sure screening property for the candidate feature selection result by the RLasso

method in 2nd step of the proposed procedure at high-dimensional scenario. Define the estimated feature index set as

$$\widehat{S} = \{i : \widehat{\theta}_{\text{RLasso},i} \neq 0, i = 1, \dots, p\}.$$

**Theorem 2.** *Consider problem (2) and let $a \doteq a_n \in (0, 1)$ be a fixed sequence such that $a \to 0$, $q \geq 1$ and $\xi > 1$ be arbitrary. Suppose that Assumptions 1, 2, 3 and 4 are satisfied. Moreover, suppose that*

$$n \geq \frac{K_1 p_0^2 \tau^4 (1 + \xi)^2 \log(p/a)}{F_q^2(\xi, X)} \tag{9}$$

*and*

$$\lambda \geq K_2 \frac{\xi + 1}{\xi - 1} \tau^2 \sqrt{\frac{\log(p/a)}{\kappa n}}, \tag{10}$$

*where $K_1, K_2, \tau$ are universal constants and $\kappa$ is the smallest eigenvalue of the correlation matrix between the true predictors $X_S = (X_{\cdot,j})_{j \in S}$. Given the beta-min condition $\min_{j \in S} |\theta_j^0| > \frac{4\xi\lambda}{\xi+1}$, then we have*

$$\lim_{n \to \infty} P(S \subset \widehat{S}) \to 1.$$

Theorem 2 indicates that when the sample size is larger, the results that the estimated relevant feature index set $\widehat{S}$ contains the true relevant feature index set $S$ satisfy the sure screening property.

**Finite-sample FDR control.**

**Theorem 3.** *Suppose the proposed model (1) satisfies all the assumptions given in the above section. Assume the statistics $W_j, 1 \leq j \leq p$, are well-defined. For any $\alpha \in (0, 1)$, the FDR of SIM-FDR satisfies*

$$\text{FDR} \leq \min_{\epsilon \geq 0} \{\alpha(1 + 4\epsilon) + P(\max_{j \in S^c} \Delta_j > \epsilon)\},$$

*where $S = \{j : \beta_j \neq 0, 1 \leq j \leq p\}$, $\Delta_j = |P(W_j > 0||W_j|, W_{-j}) - 1/2|$ and $W_{-j} = (W_1, \dots, W_p)^\top \setminus W_j$.*

This theorem holds no matter the unknown relationship between features $X$ and the response $Y$. The quantity $\Delta_j$ is seen as a measure to investigate the effect of both the asymmetry of $W_j$ and the dependence between $W_j$ and $W_{-j}$ on FDR.

**Asymptotic FDR control.** Following the proof of asymptotic FDR control [20], we need to establish the six technical assumptions for asymptotic FDR control of the proposed SIM-FDR method. These assumptions include Sure screening property for the candidate feature selection result by the 2nd step of SIM-FDR, Moments conditions, Feature matrix conditions, Estimation accuracy of the estimators $\widehat{\theta}_{\text{ROLS},i}^1$ and $\widehat{\theta}_{\text{ROLS},i}^2$ of $\theta_i^0, i \in S_0$, Signal strength, and Dependence among statistics. Assuming these assumptions hold, it is straightforward to follow the procedure of their proofs to demonstrate that our method possesses the property of asymptotic FDR control. In particular, the assumptions (Moments conditions, Feature matrix conditions, Signal strength, and Dependence) can be specified or designed, and the Assumption of "Estimation accuracy" can be achieved based on estimation consistency provided by Theorem 1 (since an estimator with asymptotic normal property must be consistent). The assumption of "Sure screening property" can be ensured by Theorem 2. Thus the asymptotic FDR control is easily proved.

## Simulation analysis results

In order to evaluate feature selection performance of the proposed method (SIM-FDR), we consider sample size $n = 250$ and 100 for moderate and small sample size scenarios, and the number of omics features $p = 400$. All simulation settings are replicated 100 times.

## Competing methods

Five methods are considered for comparisons with SIM-FDR.

1. A marginal method that testing one omics feature at a time followed by Benjamini and Hochberg (BH) correction [15], denoting "BH" method.
2. The original model-X knockoff FDR controlled feature selection method [16], denoted "MXKF" method. MXKF method uses based on high-dimensional joint linear regression model approach to analyze continuous response data. It can be implemented by using the R package *knockoffs*.
3. The regular rank-lasso method defined in (2) with the tuning parameter selected by cross-validation, denoting "Rlasso-cv" method.
4. Following the work of Rejchel et al. [32], the adaptive rank-lasso method is defined as the following formula (11),

$$\widehat{\theta}_{\text{RLasso-adaptive}} = \arg\min_{\theta}\Big[\frac{1}{2n}||Y^R - X\theta||_2^2 + \lambda_a \sum_{j=1}^{p} w_j|\theta_j|\Big], \tag{11}$$

with $\lambda_a = 2\lambda_{rl}$ and $\lambda_{rl} = 0.3\sqrt{\frac{\log(p)}{n}}$, and weights

$$w_j = \begin{cases} \frac{0.1\lambda_{rl}}{|\widehat{\theta}_{rl,j}|}, & |\widehat{\theta}_{rl,j}| > 0.1\lambda_{rl}; \\ |\widehat{\theta}_{rl,j}|^{-1}, & \text{otherwise}, \end{cases}$$

where

$$\widehat{\theta}_{rl} = \arg\min_{\theta}\Big[\frac{1}{2n}||Y^R - X\theta||_2^2 + \lambda_{rl}\sum_{j=1}^{p}|\theta_j|\Big]. \tag{12}$$

If $\widehat{\theta}_{rl,j} = 0$, then the $j$-th explanatory variable is removed from the list of predictors before running adaptive rank-lasso (11). This method is denoted as "Rlasso-adaptive".

5. Following the work of Rejchel et al. [32], the threshold rank-lasso is defined as: the tuning parameter for rank-lasso is selected by cross-validation and the threshold is selected in such a way that the number of selected predictors coincides with the number of predictors selected by adaptive rank-lasso (the above method). This method is denoted as "Rlasso-threshold".

## Generating omics features

We simulate the $n \times p$ feature matrix $X$ by the multivariate normal distribution $N_p(\mu, \Sigma)$ with $\mu = (\mu_1, \ldots, \mu_p)^\top$ and $\mu_i = 0$. To evaluate the performance of SIM-FDR under the dependency among the features, we set the covariance matrix $\Sigma = (\Sigma_{ij})_{p \times p}$ to be the following structure.

- $\Sigma_{ij} = \sigma^{|i-j|}$ for $i \neq j, i, j = 1, \ldots, p$, where we set $\sigma = 0.5$ for moderate strength correlation level among omics features and $\Sigma_{ii} = 1$ for $i = 1, \ldots, p$.

### Simulating the response

We firstly design the regression coefficients $\beta = (\beta_1, \ldots, \beta_p)$ and then use the coefficients to generate the response. Let the location vector of nonzero values in $\beta$ be

$$V = \underbrace{(1, 2, 3, 4, 5, 10, 20, 30, 40, 50)}_{10}.$$

Then we set $\beta[V]$, which denotes the nonzero values vector of $\beta$, to be

$$\beta[V] = \underbrace{(-1.5, 1.5, -2, 2, -1, 1, -3, 3, 2, -2)}_{10}.$$

After generating $\beta$, we employ the following six different type models to simulate the outcome $Y$, where linear regression model is used to simulate linear associated relationships between the response and the omics features, single index model is used to simulate nonlinear associated relationships, and the Cauchy distribution is used to simulate the heavy-tailed distributional random errors.

- Model 1
  Linear regression model setting with the normal distributional random error: $y_i = \Sigma_{j=1}^{p} \beta_j X_{i,j} + \varepsilon_i$, where the error term $\varepsilon$ is independent of $X$ and generated from the normal distribution with location parameter being 0 and the variance parameter being $\gamma$.
- Model 2
  Linear regression model setting with the Cauchy distributional random error: $y_i = \Sigma_{j=1}^{p} \beta_j X_{i,j} + \varepsilon_i$, where the error term $\varepsilon$ is independent of $X$ and generated from the Cauchy distribution with location parameter being 0 and the scale parameter being $\gamma$.
- Model 3
  Single index model with the normal distributional random error: $y_i = \exp\left(4 + \Sigma_{j=1}^{p} \beta_j X_{i,j}\right) + \varepsilon_i$, where the error term $\varepsilon$ is independent of $X$ and generated from the standard normal distribution with location parameter being 0 and the scale parameter being $\gamma$.
- Model 4
  Single index model setting with the Cauchy distributional random error: $y_i = \exp\left(4 + \Sigma_{j=1}^{p} \beta_j X_{i,j}\right) + \varepsilon_i$, where the error term $\varepsilon$ is independent of $X$ and generated from the Cauchy distribution with location parameter being 0 and the scale parameter being $\gamma$.
- Model 5: Double-index model.
  Let the location vector of nonzero values in $p \times 1$-dimensional $\beta_1$ be

$$V_1 = \underbrace{(1, 2, 3, 4, 5)}_{5},$$

and the location vector of nonzero values in $p \times 1$-dimensional $\beta_2$ be

$$V_2 = \underbrace{(10, 20, 30, 40, 50)}_{5}.$$

Then we set $\beta_1[V_1]$ and $\beta_2[V_2]$ to be

$$\underbrace{(-1.5, 1.5, -2, 2, -1)}_{5}$$

and

$$(\underbrace{1, -3, 3, 2, -2}_{5}),$$

respectively. Double-index model setting with the Cauchy distributional random error is considered:

$$y_i = \Sigma_{j=1}^{p} \beta_{1,j} X_{i,j} + \exp(\Sigma_{j=1}^{p} \beta_{2,j} X_{i,j}) + \varepsilon_i,$$

where the error term $\varepsilon$ is independent of $X$ and generated from the Cauchy distribution with location parameter being 0 and the scale parameter being $\gamma$.

- Model 6: Multi-index model.

Let the location vector of nonzero values in $p \times 1$-dimensional $\beta_1$ be

$$V_1 = (\underbrace{1, 2, 3}_{3}),$$

the location vector of nonzero values in $p \times 1$-dimensional $\beta_2$ be

$$V_2 = (\underbrace{4, 5, 10}_{3}),$$

and the location vector of nonzero values in $p \times 1$-dimensional $\beta_3$ be

$$V_3 = (\underbrace{20, 30, 40, 50}_{4}).$$

Then we set $\beta_1[V_1]$, $\beta_2[V_2]$ and $\beta_3[V_3]$ to be

$$(\underbrace{-1.5, 1.5, -2}_{3}),$$

$$(\underbrace{2, -1, 1}_{3}),$$

and

$$(\underbrace{-3, 3, 2, -2}_{4}),$$

respectively. Multi-index model setting with the Cauchy distributional random error is considered:

$$y_i = \frac{\Sigma_{j=1}^{p} \beta_{1,j} X_{i,j}}{\exp(\Sigma_{j=1}^{p} \beta_{2,j} X_{i,j})} + \exp(\Sigma_{j=1}^{p} \beta_{3,j} X_{i,j}) + \varepsilon_i,$$

where the error term $\varepsilon$ is independent of $X$ and generated from the Cauchy distribution with location parameter being 0 and the scale parameter being $\gamma$.

In order to consider the different strength association levels between the features and the response, we vary the signal noise ratio (SNR), defined as $\frac{\text{Var}(E(Y|Z))}{\gamma}$ for models 1 and 2, where the scale or variance parameter can be set for models 3, 4, 5 and 6 by the formula $\gamma = \frac{\text{Var}(E(Y|Z))}{\text{SNR}}$.

## Methods settings and comparison measurements

The MXKF method places the burden of knowledge on knowing the complete conditional distribution of $X$, and there is no algorithm that can generate model-X knockoffs for general distributions efficiently [18]. Therefore, we utilize the default design used previously [17] in this simulation. For SIM-FDR methods, the optimal $\lambda$ used in the rank-based lasso are determined through 10-fold cross-validation. For BH method, we test the association between the outcome and each omics feature marginally and apply the BH procedure to these marginal $p$-values to identify significant features. In addition, we set $\gamma = 2/3$ and $B = 10$ for SIM-FDR in simulation analysis.

Given nominal FDR levels $\alpha = 0.05, 0.1$, based on 100 simulated data sets, we use empirical FDR and empirical Power, defined as

$$\widehat{\text{FDR}} = \frac{1}{100} \sum_{i=1}^{100} \left[ \frac{|\{j : I_j = 0 \text{ and } j \in S_i\}|}{|S_i| \vee 1} \right];$$

$$\widehat{\text{Power}} = \frac{1}{100} \sum_{i=1}^{100} \left[ \frac{|\{j : I_j \neq 0 \text{ and } j \in S_i\}|}{|S^*|} \right],$$

to measure the feature selection performance of different methods. In addition, the Matthews correlation coefficient (MCC) is employed to evaluate the results of feature selection. MCC measures the overall accuracy of selection for true positives (TP), false negatives (FN), true negatives (TN), and false positives (FP), with a larger value indicating overall better selection [43]. The definition of empirical MCC is

$$\widehat{\text{MCC}} = \frac{1}{100} \sum_{i=1}^{100} \left[ \frac{\widehat{\text{TP}}_i \times \widehat{\text{TN}}_i - \widehat{\text{FP}}_i \times \widehat{\text{FN}}_i}{\sqrt{(\widehat{\text{TP}}_i + \widehat{\text{FP}}_i) \times (\widehat{\text{TP}}_i + \widehat{\text{FN}}_i) \times (\widehat{\text{TN}}_i + \widehat{\text{FP}}_i) \times (\widehat{\text{TN}}_i + \widehat{\text{FN}}_i)}} \right]$$

with

$$\widehat{\text{TP}}_i = |\{j : I_j \neq 0 \text{ and } j \in S_i\}|,$$

$$\widehat{\text{TN}}_i = |\{j : I_j = 0 \text{ and } j \notin S_i\}|,$$

$$\widehat{\text{FP}}_i = |\{j : I_j = 0 \text{ and } j \in S_i\}|,$$

$$\widehat{\text{FN}}_i = |\{j : I_j \neq 0 \text{ and } j \notin S_i\}|,$$

where $I_j = 0$ indicates that the $j$-th omics feature is not associated with the response, and $I_j \neq 0$ indicates that the $j$-th omics feature is truly associated with the response, $S^* = \{j : I_j \neq 0, j = 1, \dots, p\}$ denotes the indices set of omics features truly associated with the response, $|S^*|$ denotes the number of omics features truly associated with the response, and $S_i$ denotes the indices set of the selected omics feature using the $i$-th data set.

## Results for moderate sample size scenario (n = 250)

The FDR performance of all the methods is similar for models 1 and 2, both FDR and power performances of all the methods are nearly the same for models 3 and 4, and the FDR and power performances of all the methods are nearly the same for models 5 and 6. Consequently, we present the analysis results for models 1 and 2, models 3 and 4, and models 5 and 6, respectively.

**Results for models 1 and 2.**

- FDR performance.
  For models 1 and 2, as shown in Figs 1 and 2, the proposed SIM-FDR method effectively controls the actual FDRs at the specified levels for all simulation scenarios. BH and Rlasso-cv methods exhibit significantly higher actual FDRs across all simulation scenarios and fails to control the FDR as expected. The MXKF method has significantly higher or lightly higher actual FDRs than the specified FDR levels for all simulation scenarios. The actual FDRs of both Rlasso-adaptive and Rlasso-threshold methods are nearly zero.
- Power performance.
  For model 1, the MXKF method demonstrates nearly identical power to our SIM-FDR method but at the expense of higher actual FDRs. While for model 2 scenario with the Cauchy distributional errors, SIM-FDR substantially outperforms the MXKF method with the power improvement approaching about 0.36 for some scenarios, and significantly dominates Rlasso-adaptive and Rlasso-threshold methods with the power improvement approaching about 0.62 for some scenarios. BH and Rlasso-cv method possesses too high actual FDRs for these two models so that their performance becomes inconsequential.

**Results for models 3 and 4.** From Figs 3 and 4, the results for models 3 and 4 demonstrate the same performance for all the methods. Here, we present their results together. For models 3 and 4, SIM-FDR, Rlasso-adaptive and Rlasso-threshold methods effectively control the actual FDRs to the specified levels across all scenarios. In contrast, the MXKF and Rlasso-cv methods exhibit much higher actual FDRs for almost of all scenarios and do not achieve the expected level of FDR control. In addition, SIM-FDR demonstrates significantly better power performance than MXKF, Rlasso-adaptive and Rlasso-threshold methods across all scenarios, with power improvement being substantial and approaching about 0.7 for some scenarios.

**Results for models 5 and 6.** From Figs 5 and 6, the results for models 5 and 6 demonstrate SIM-FDR, Rlasso-adaptive and Rlasso-threshold methods effectively control the actual FDRs to the specified levels across all scenarios. BH method only can control FDRs for some scenarios. While the MXKF and Rlasso-cv methods own much higher actual FDRs for all scenarios and can not achieve the expected level of FDR control. For power performance, SIM-FDR significantly outperforms MXKF, BH, Rlasso-adaptive and Rlasso-threshold methods across all scenarios, with power improvement being substantial and approaching about 0.62 for some simulation scenarios. However, even though the power of Rlasso-cv is highest, lacking effective FDR control makes its performance to be inconsequential.

## Results for small sample size scenario (n = 100)

The performance of all the methods is similar for models 1 and 2, and models 3 and 4. The analysis results for models 1 and 2, and models 3 and 4 are shown, respectively.

**Results for models 1 and 2.** As shown in Figs 7 and 8, the SIM-FDR, Rlasso-adaptive, and Rlasso-threshold methods effectively control the actual FDRs at the specified levels for all simulation scenarios. However, the BH and MXKF methods exhibit significantly higher actual FDRs for most simulation scenarios and fail to control the actual FDRs at the desired levels. Regarding these two models, as illustrated in Figs 7 and 8, the MXKF method shows slightly higher power than SIM-FDR, albeit at the cost of higher actual FDRs. In terms of

## (A) Model 1, moderate sample                    (B) Model 1, moderate sample

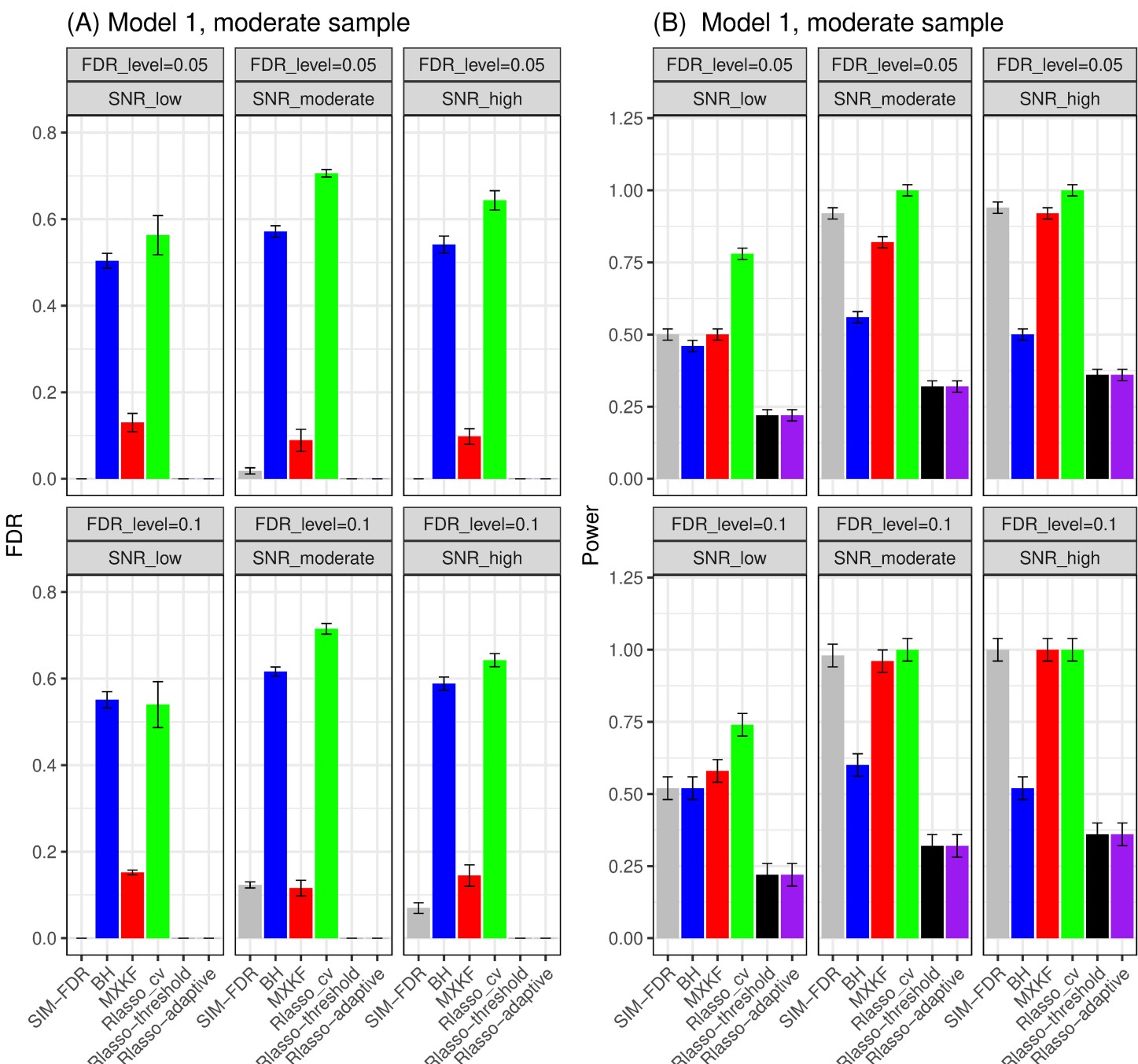

**Fig 1. Results for model 1 at moderate sample size case (n = 250)** FDRs given in left column and powers given in right column are averaged over 100 replications, and their standard deviations (sd) are given on the top of the histogram.

power performance, our method SIM-FDR consistently outperforms the Rlasso-adaptive and Rlasso-threshold methods across all simulation scenarios.

**Results for models 3 and 4.** From Figs 9 and 10, it can be observed that for models 3 and 4, the SIM-FDR, Rlasso-adaptive, and Rlasso-threshold methods effectively control the actual false discovery rates (FDRs) to the specified levels in all simulation scenarios. The Benjamini-Hochberg (BH) method shows success in controlling FDR for certain scenarios, while the

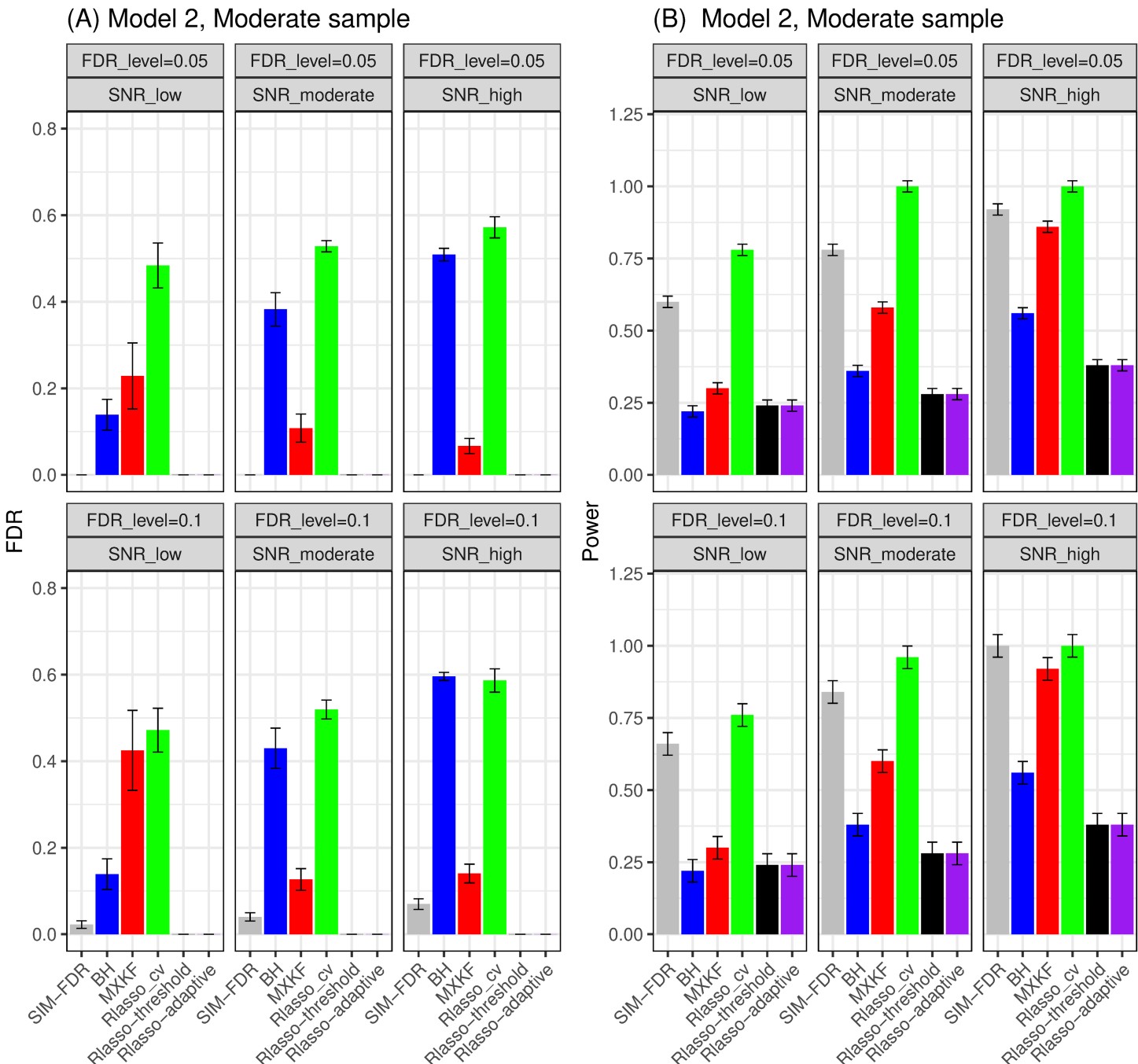

**Fig 2. Results for model 2 at moderate sample size case (n = 250)** FDRs given in left column and powers given in right column are averaged over 100 replications, and their standard deviations (sd) are given on the top of the histogram.

MXKF method exhibits significantly higher actual FDRs and lacks the ability to control FDRs in all simulation scenarios. In terms of power performance, SIM-FDR demonstrates superiority over the BH, MXKF, Rlasso-adaptive, and Rlasso-threshold methods across all simulation scenarios, with a substantial power improvement of approximately 0.30 in some scenarios.

**Results for model 5.** Since the statistical power of most methods is close to zero in scenarios with small sample sizes for model 6, we will only focus on the results of model 5. From

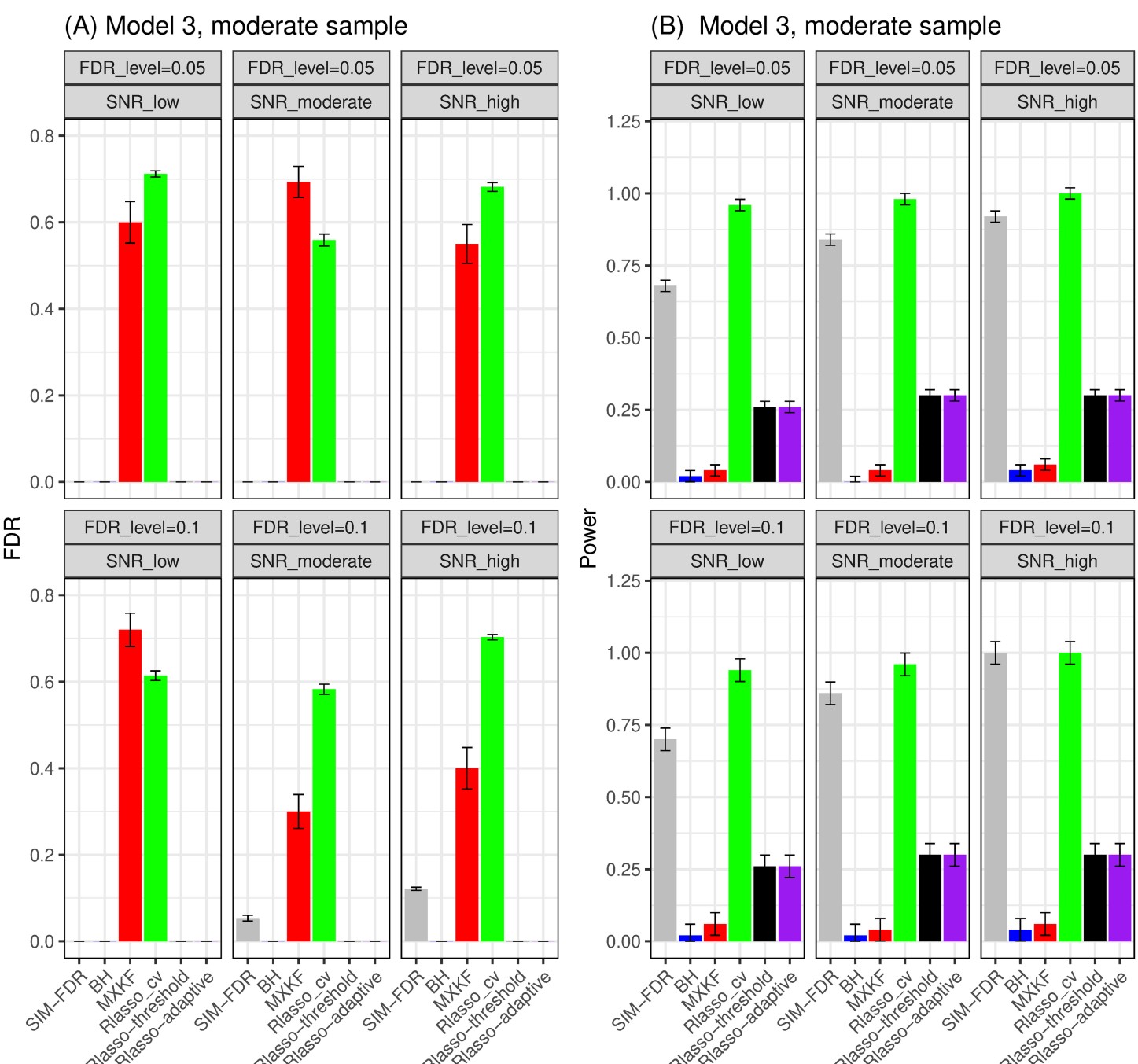

**Fig 3. Results for model 3 at moderate sample size case (n = 250)** FDRs given in left column and powers given in right column are averaged over 100 replications, and their standard deviations (sd) are given on the top of the histogram.

Fig 11, the results show that BH, SIM-FDR, Rlasso-adaptive, and Rlasso-threshold methods effectively control the actual false discovery rates (FDRs) to the specified levels across all scenarios in the small sample simulation scenario. However, the MXKF and Rlasso-cv methods have much higher actual FDRs for all scenarios and fail to achieve the expected level of FDR

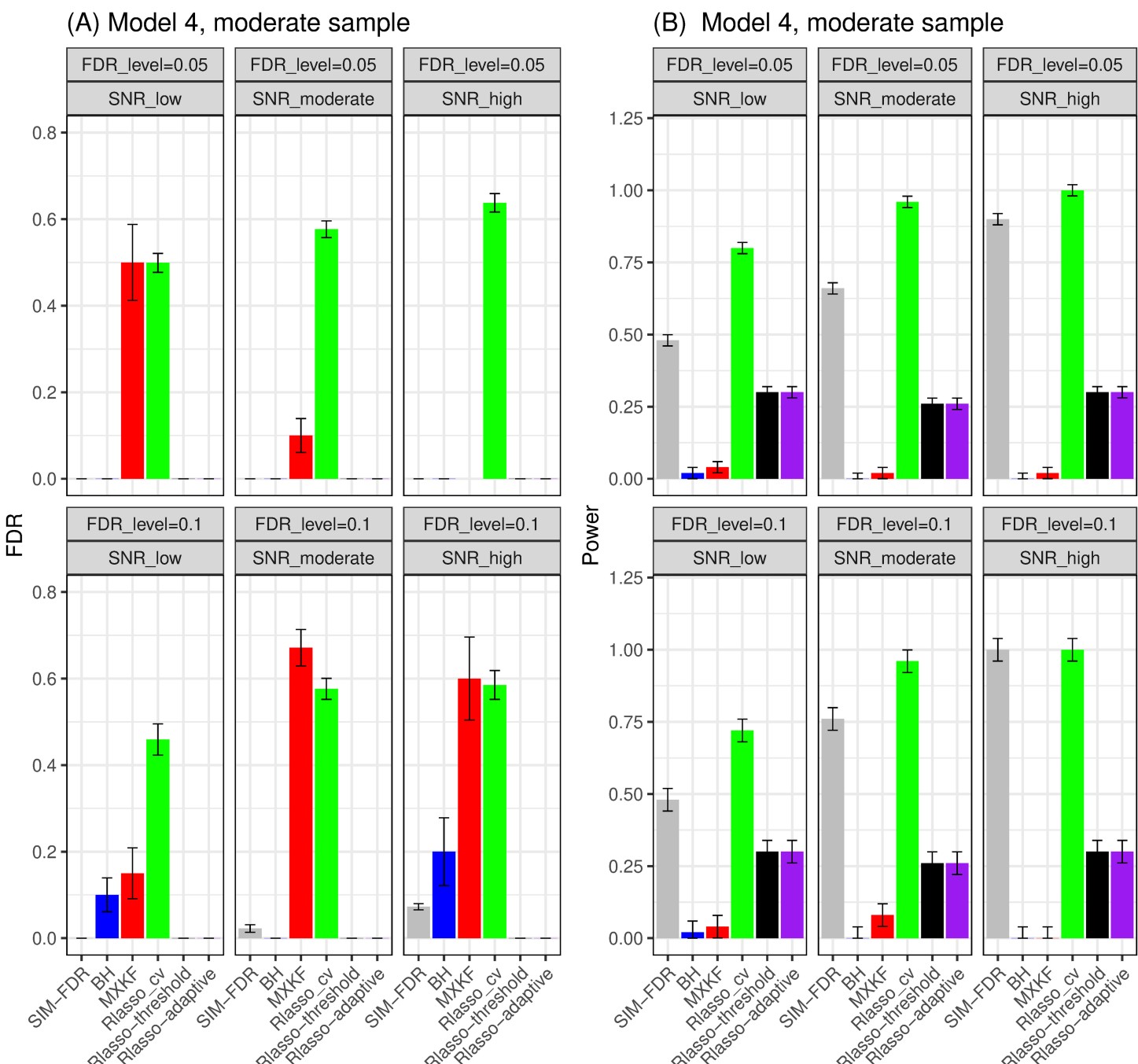

**Fig 4. Results for model 4 at moderate sample size case (n = 250)** FDRs given in left column and powers given in right column are averaged over 100 replications, and their standard deviations (sd) are given on the top of the histogram.

control. In terms of power performance, SIM-FDR outperforms MXKF, BH, Rlasso-adaptive, and Rlasso-threshold methods across all simulation scenarios.

**The simulation results evaluated via the Matthews correlation coefficient (MCC).**
All MCC results are presented in Figs 12, 13, 14, 15, 16, 17, 18, 19, 20, 21, and 22. Notably, Figs 12, 13, 14, 15, 21, and 22 demonstrate that our method, SIM-FDR, achieves the highest

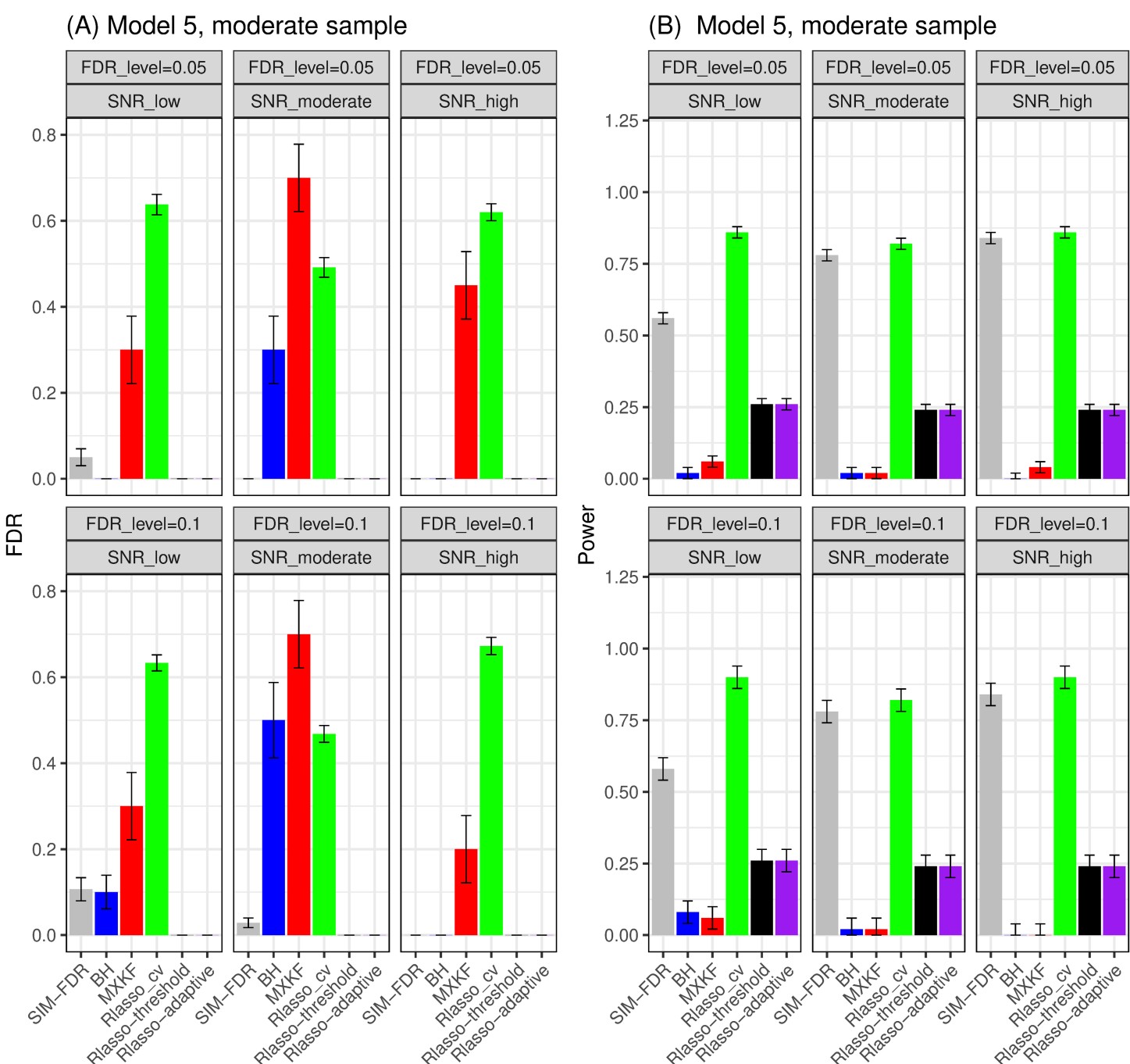

**Fig 5. Results for model 5 at moderate sample size case (n = 250)** FDRs given in left column and powers given in right column are averaged over 100 replications, and their standard deviations (sd) are given on the top of the histogram.

MCC values across 36 scenarios under moderate sample size ($n$ = 250), indicating superior feature selection performance for moderate to large sample sizes. For small sample size ($n$ = 100), SIM-FDR consistently outperforms competing methods for 12 scenarios depicted in Figs 18 and 20. While in Figs 16, 17, and 19, SIM-FDR dominates or significantly outperforms other methods in 10 out of 18 scenarios, and only falls behind the MXKF method in the rest

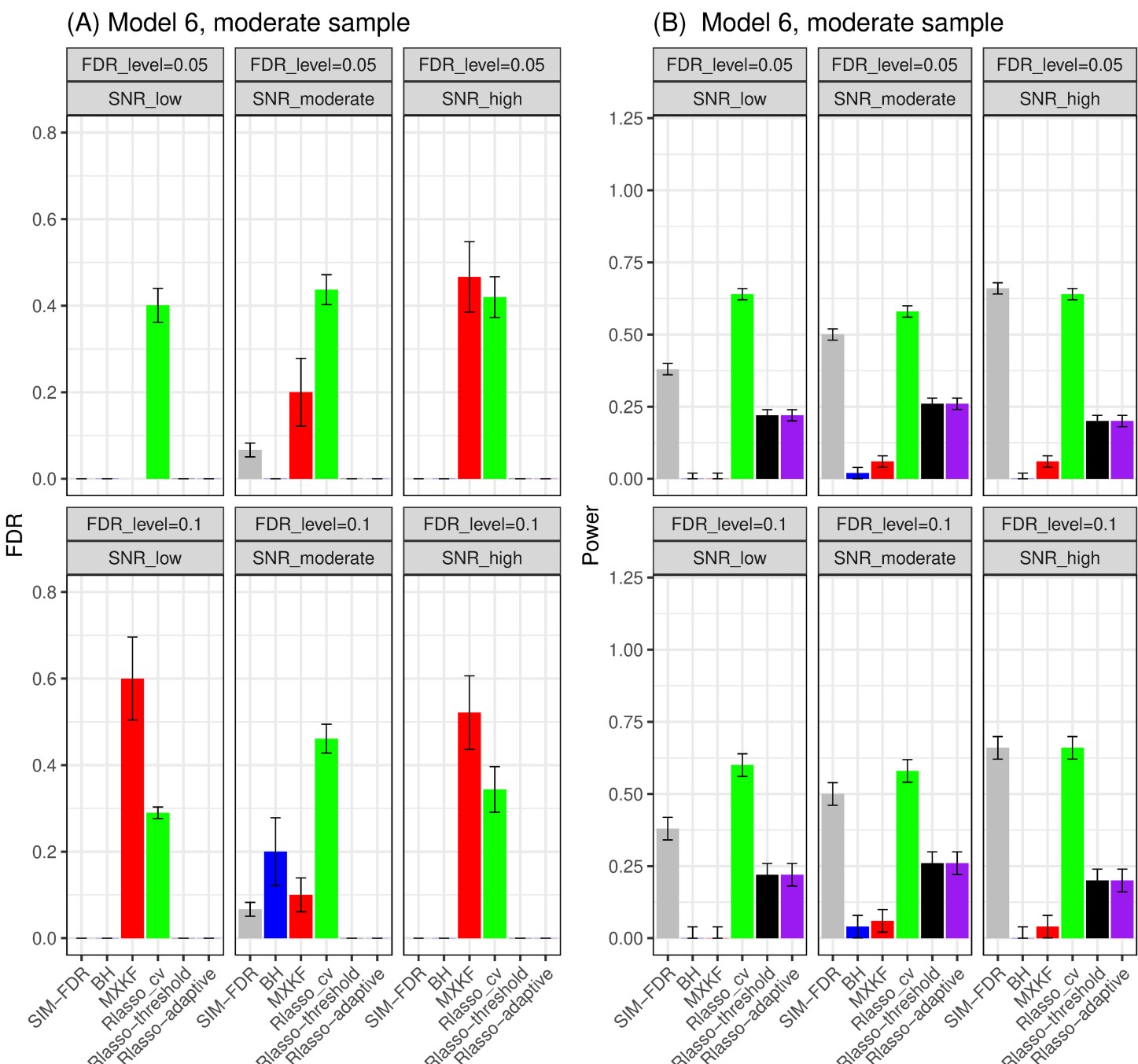

**Fig 6. Results for model 6 at moderate sample size case (n = 250)** FDRs given in left column and powers given in right column are averaged over 100 replications, and their standard deviations (sd) are given on the top of the histogram.

8 scenarios. In a word, our method outperforms the other methods for 58 scenarios among the total 66 scenarios and ranks second in the remaining 8 scenarios. However, as shown in Figs 7, 8, and 10, MXKF exhibits substantially higher actual FDRs in this 8 scenarios, failing to control false discoveries and yielding excessive false positives. While MCC serves as a comprehensive evaluation metric, this paper primarily focuses on FDR and Power performance of

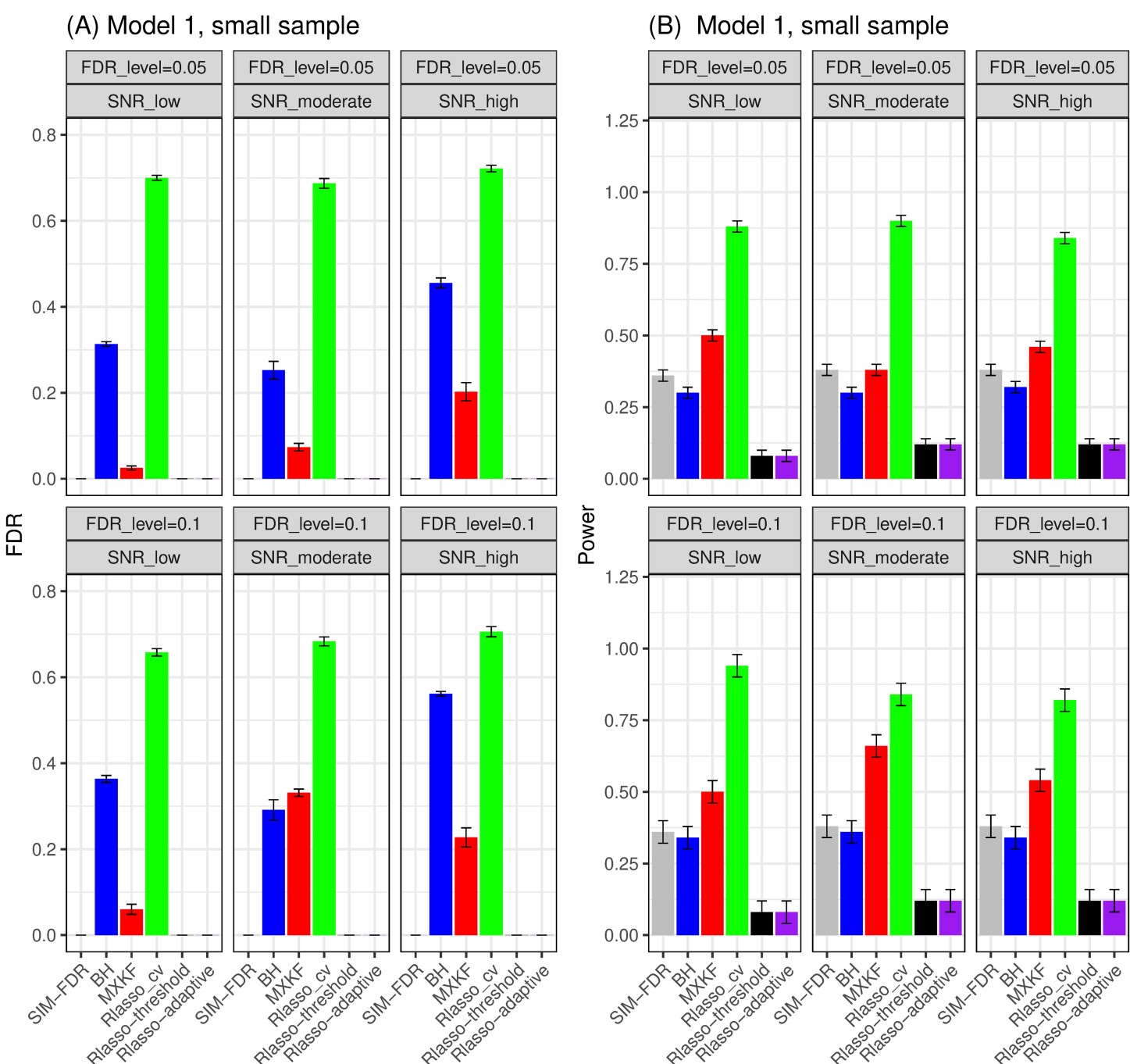

**Fig 7. Results for model 1 at small sample size case (n = 100)** FDRs given in left column and powers given in right column are averaged over 100 replications, and their standard deviations (sd) are given on the top of the histogram.

all the methods. Consequently, MXKF's marginally better MCC in the remaining 8 scenarios may lack practical significance for real-world data analysis.

**Conclusions of simulating analysis results.** Reviewing the results from almost of all the scenarios, the actual FDRs of SIM-FDR are far below the actual FDRs of the other methods. Thence, there should be no need to verify that our procedure has higher power at the price of

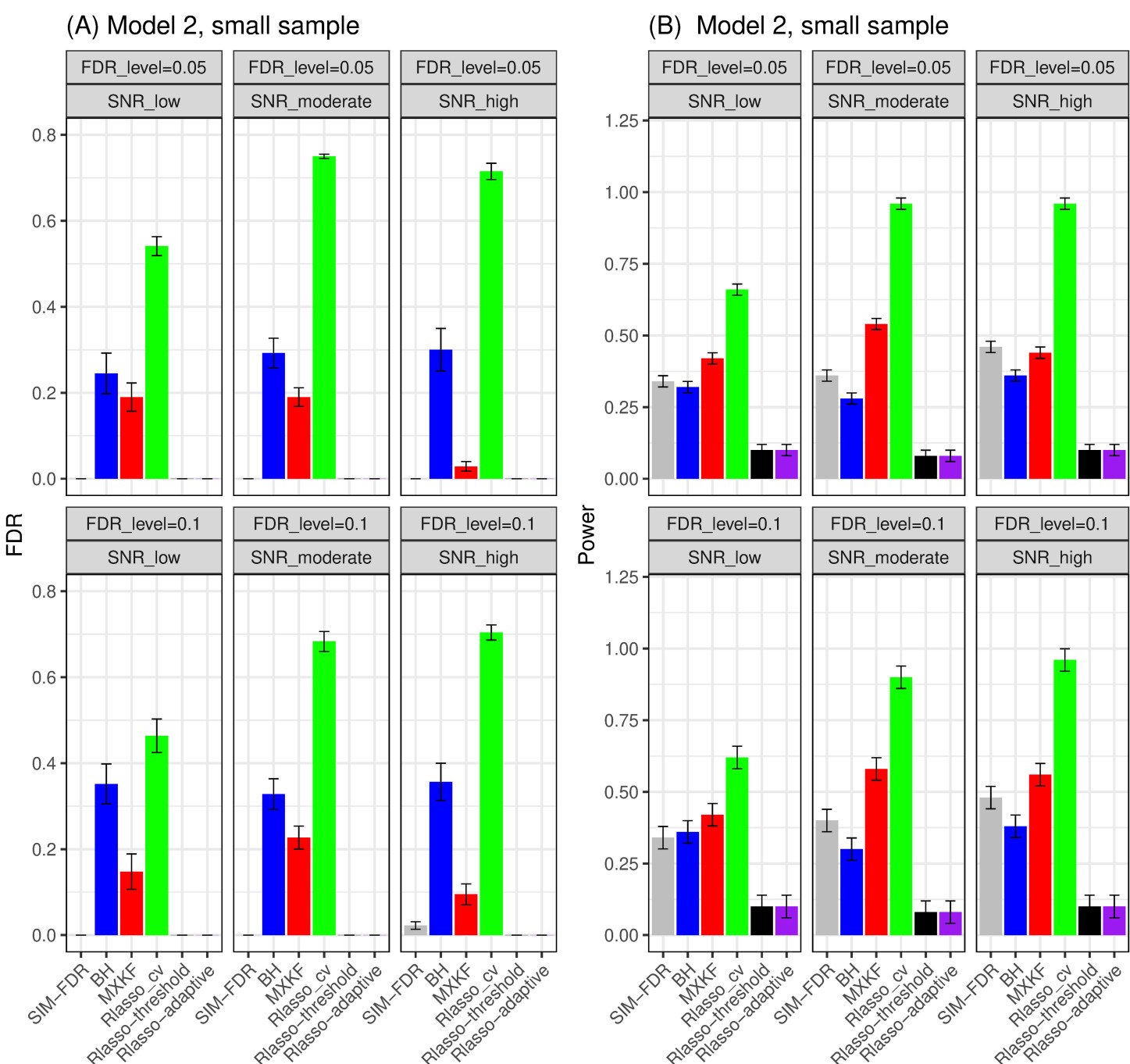

**Fig 8. Results for model 2 at small sample size case (n = 100)** FDRs given in left column and powers given in right column are averaged over 100 replications, and their standard deviations (sd) are given on the top of the histogram.

a higher FDR level. To summarise, compared to SIM-FDR method, other existing methods may either be underpowered or render inappropriate results by having an inflated FDR than the nominal FDR threshold. Especially, the proposed method owns better feature selection performance for moderate to large sample sizes. Such results indicate the proposed SIM-FDR owns robust performance for all the scenarios.

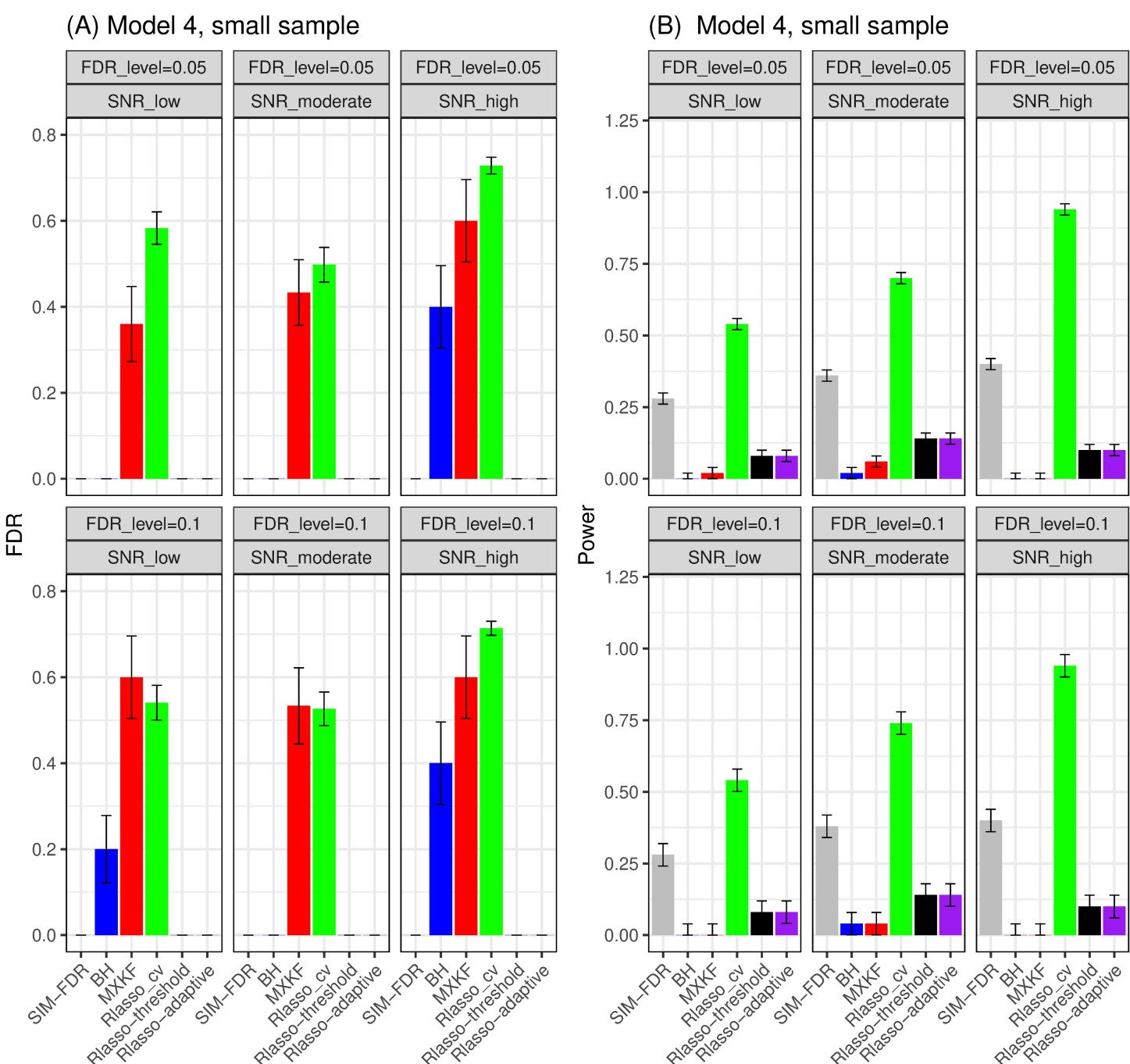

**Fig 9. Results for model 3 at small sample size case (n = 100)** FDRs given in left column and powers given in right column are averaged over 100 replications, and their standard deviations (sd) are given on the top of the histogram.

## Real data analysis results

### Ocean microbiome data

Integrative marine data collection efforts such as Tara Oceans [44] or the Simons CMAP provides the means to investigate ocean ecosystems on a global scale. This data set contains

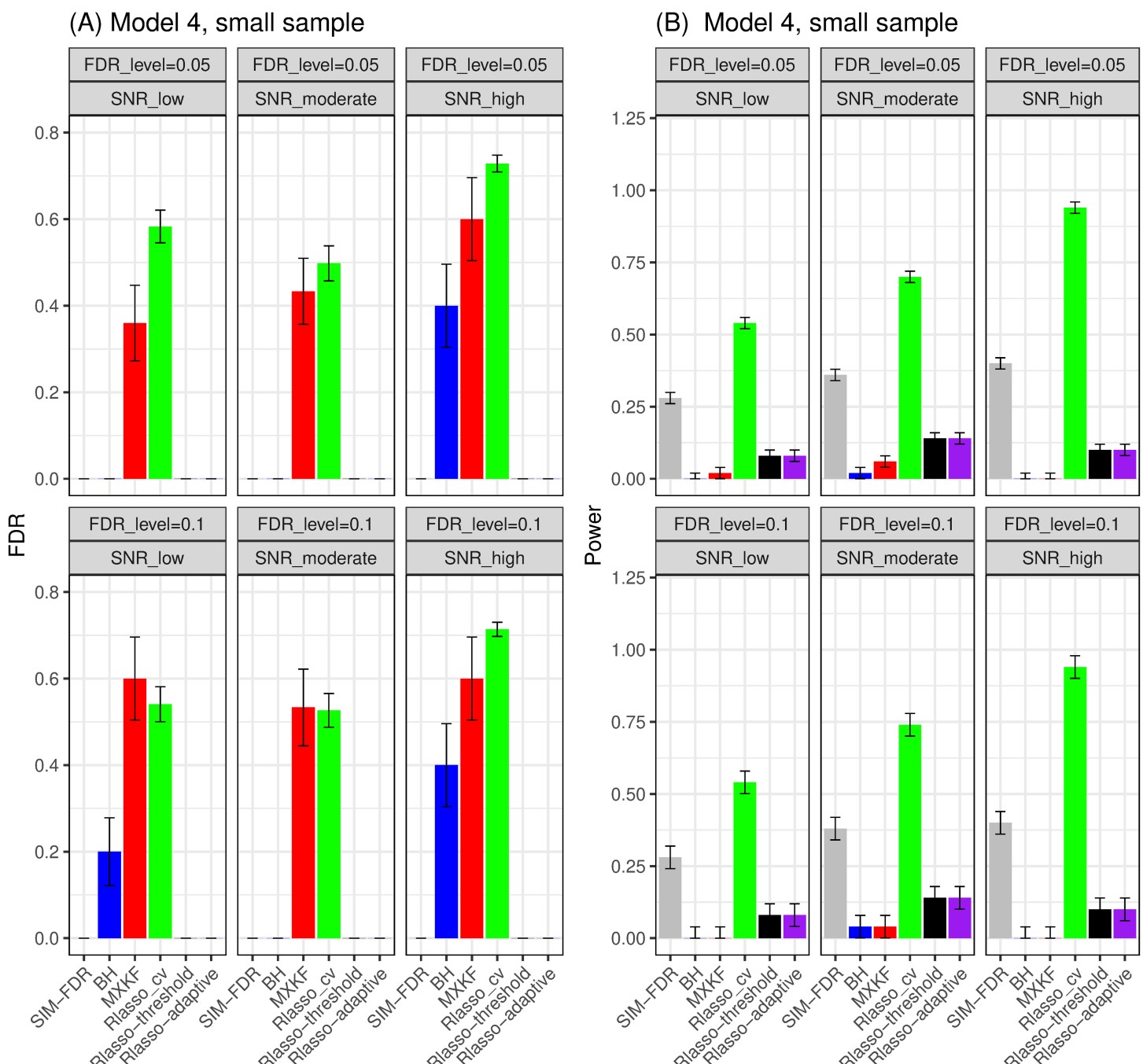

**Fig 10. Results for model 4 at small sample size case (n = 100)** FDRs given in left column and powers given in right column are averaged over 100 replications, and their standard deviations (sd) are given on the top of the histogram.

$p$ = 35651 miTAG OTUs [45] observed on $n$ = 136 samples. Using Tara's environmental and microbial survey of ocean surface water [45], we apply all the methods to identify miTAG OTUs (omics features) associated with some environmental covariates. Especially, salinity is thought to be an important environmental factor in marine microbial ecosystems, thus

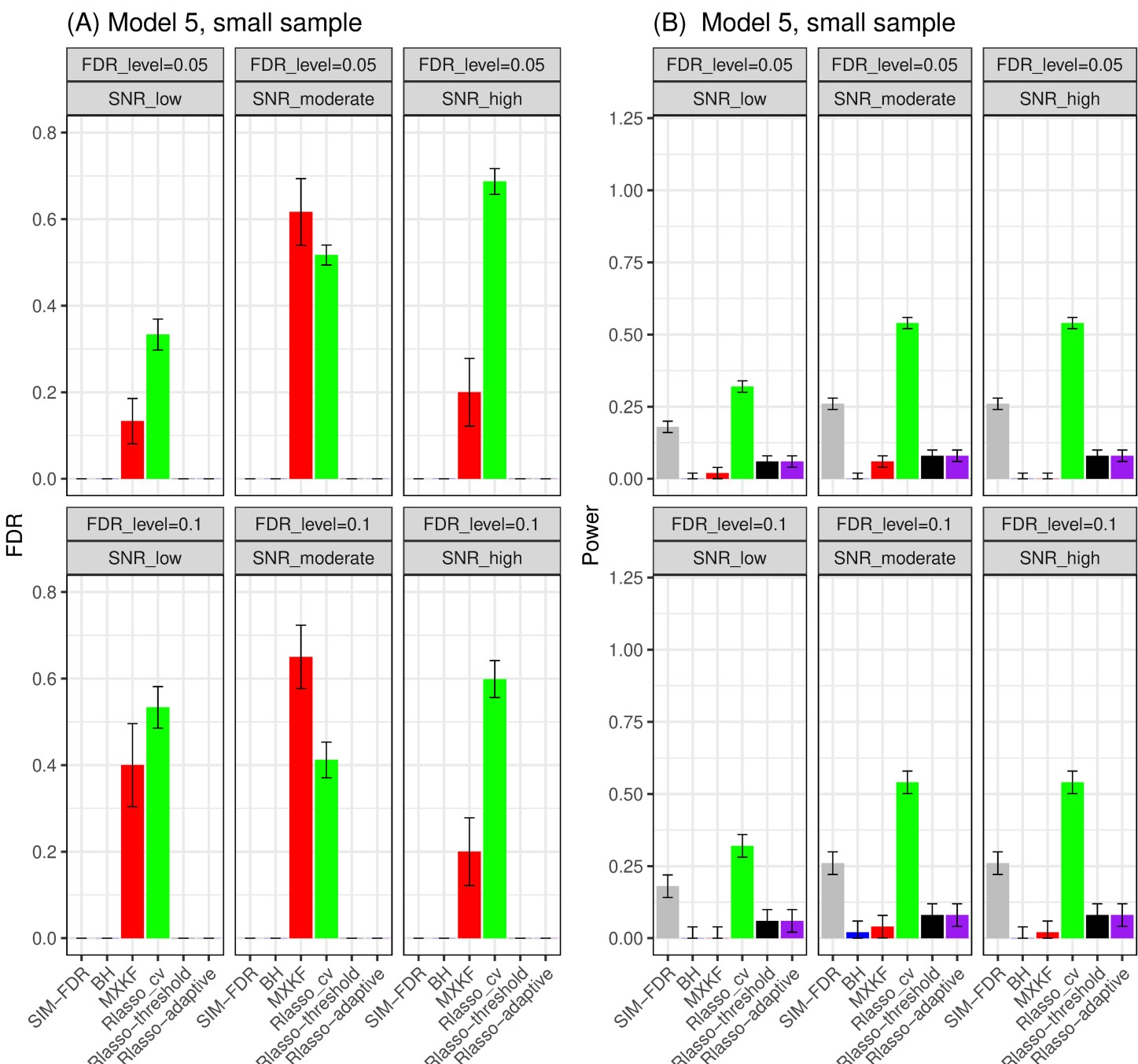

**Fig 11. Results for model 5 at small sample size case (n = 100)** FDRs given in left column and powers given in right column are averaged over 100 replications, and their standard deviations (sd) are given on the top of the histogram.

we aimed to identify the miTAG OTUs (omics features) more robustly associated with the response of interest "marine salinity".

Before applying all the methods, we conducted a series of preprocessing steps to make the Tara data more amenable to the proposed method. Firstly, following the work of Sunagawa et al. [46], we calculated read sum of all the 35651 miTAG OTUs (omics features) and

## Model 1, moderate sample

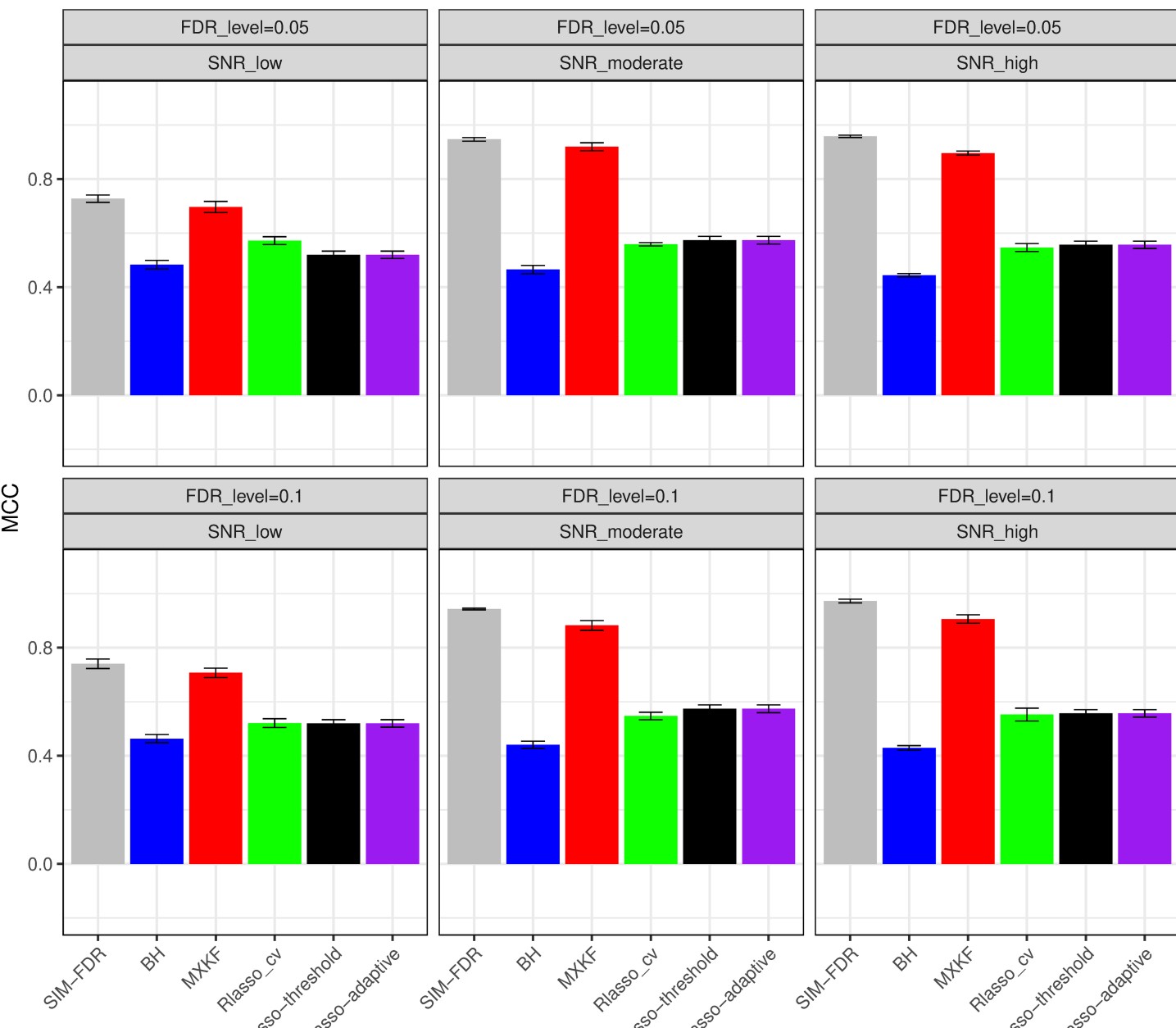

**Fig 12. Results for model 1 at moderate sample size case (n = 250)** MCCs are averaged over 100 replications, and their standard deviations (sd) are given on the top of the histogram.

removed low-abundance OTUs with read sum less than 10000 reads per sample. We further retained OTUs that appeared in at least 14 samples, resulting in a new OTUs matrix of dimension $n = 136$ and $p = 1015$. Secondly, we normalized OTU raw read counts into composition data with the sum of each row being one. Thirdly, we transformed the composition data using log function and took the log-transformed data as the omics features. Following the analysis of simulated data, we set the same setting $\gamma = 2/3$ for SIM-FDR while let $B$ be 30

## Model 2, moderate sample

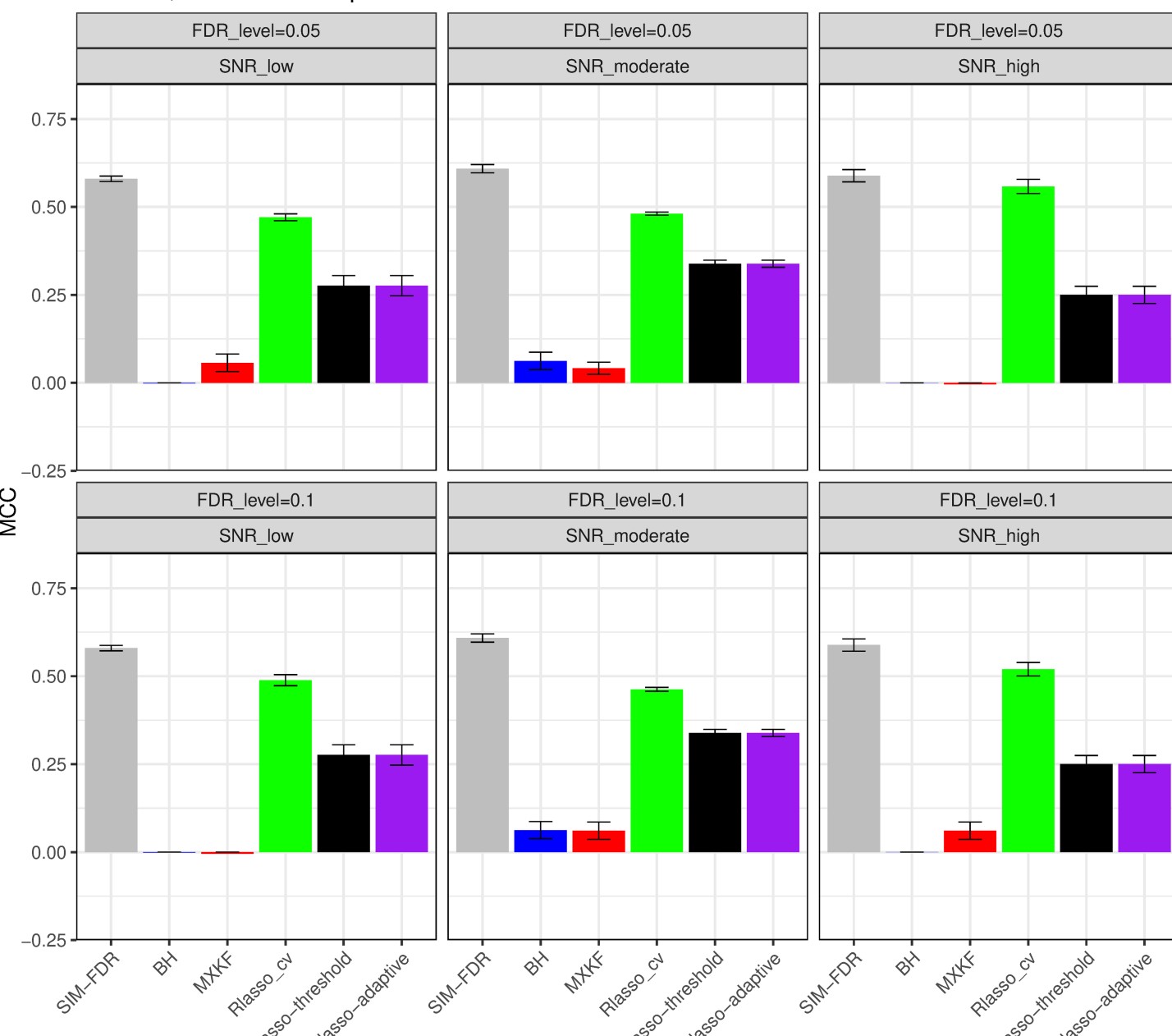

**Fig 13. Results for model 2 at moderate sample size case (n = 250)** MCCs are averaged over 100 replications, and their standard deviations (sd) are given on the top of the histogram.

to stabilize the analysis results of SIM-FDR. We varied nominal FDR levels from 0 to 0.20 for the real data analysis. The collected raw Tara data set and the preprocessed Tara data set can be available in S2 Dataset and S3 Dataset (Supporting information section), respectively.

The results were presented in Table 1. It is observed that the number of taxa identified by BH and Rlasso-cv methods exceed that of SIM-FDR, MXKF, Rlasso-threshold and Rlasso-adaptive methods, yet MXKF did not identify any taxa fetures for all the FDR levels.

## Model 3, moderate sample

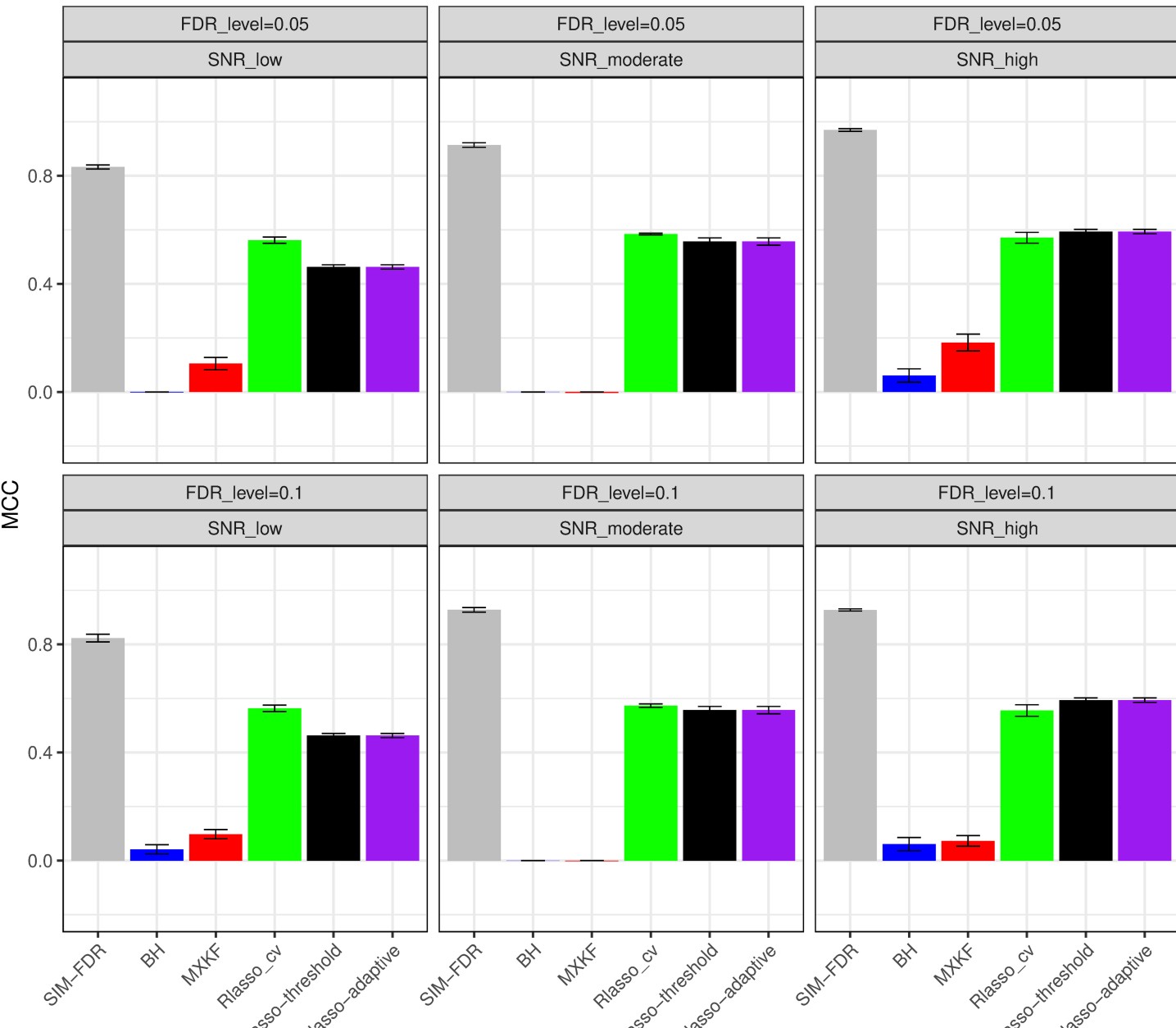

**Fig 14. Results for model 3 at moderate sample size case (n = 250)** MCCs are averaged over 100 replications, and their standard deviations (sd) are given on the top of the histogram.

This may align with the simulation results from many scenarios. For example, the scenarios may be model 1 or model 2 with small sample scenario (Figs 7 and 8), giving the sample size of $n$ = 136 and the number of genes $p$ = 1015. The Figs 7 and 8 show that the BH and Rlasso-cv methods exhibit higher actual FDRs and fails to control FDR at the given nominal levels. Therefore, the results in Table 1 may indicate that BH and Rlasso-cv methods may yield

## Model 4, moderate sample

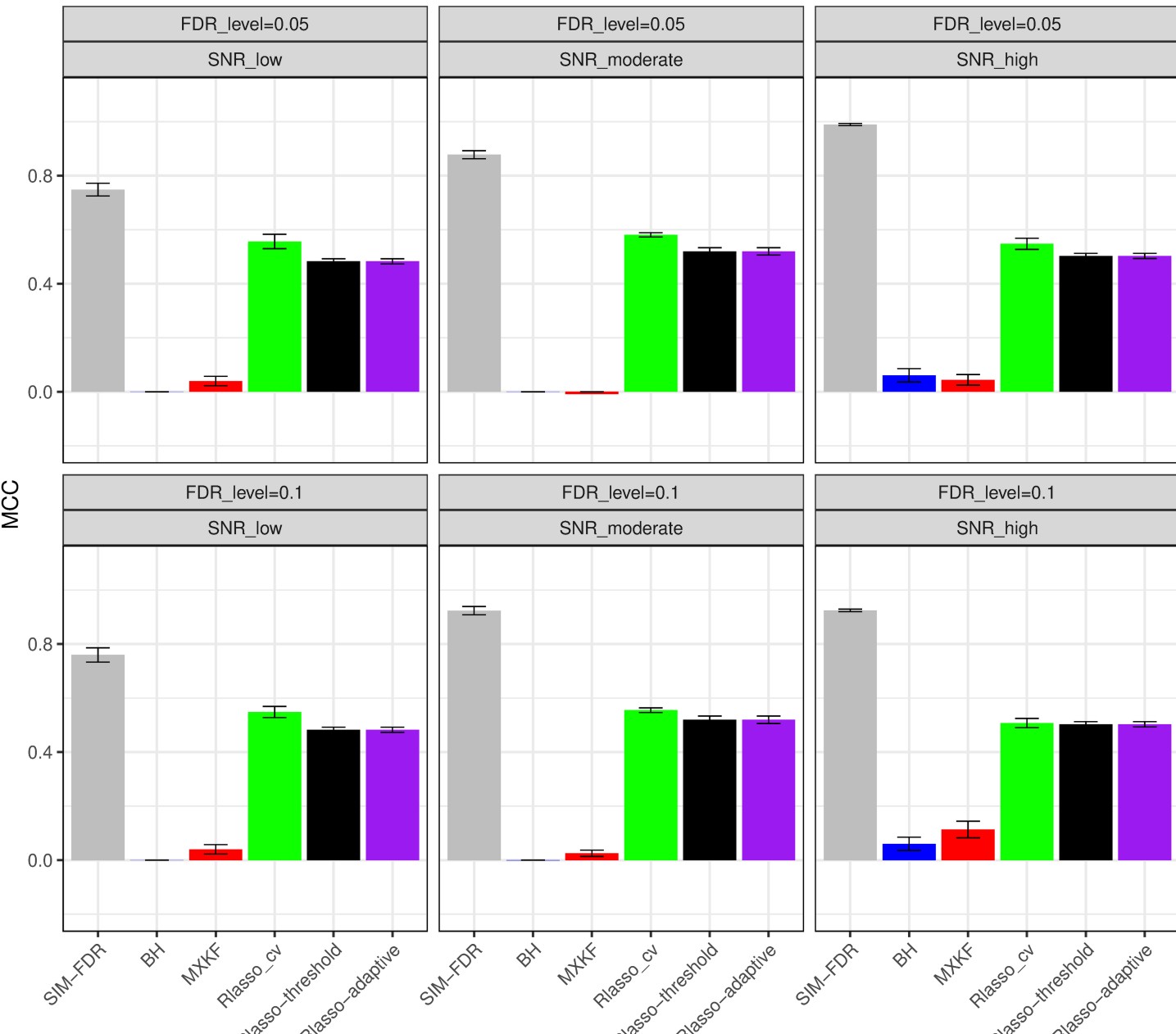

**Fig 15. Results for model 4 at moderate sample size case (n = 250)** MCCs are averaged over 100 replications, and their standard deviations (sd) are given on the top of the histogram.

more false discoveries, while SIM-FDR, Rlasso-threshold and Rlasso-adaptive methods may provide fewer but more precise taxa selection results.

However, we need to further validate that SIM-FDR should yield more precise taxon selection compared to BH and Rlasso-cv methods. To investigate the feature selection performance of our proposed method, we introduce simulated variables as inactive or non-association taxa. In detail, 500 noise taxa $Z_1$ are from $N(0,1)$ and another 500 noise taxa $Z_2$ are from $t(3)$,

## Model 1, small sample

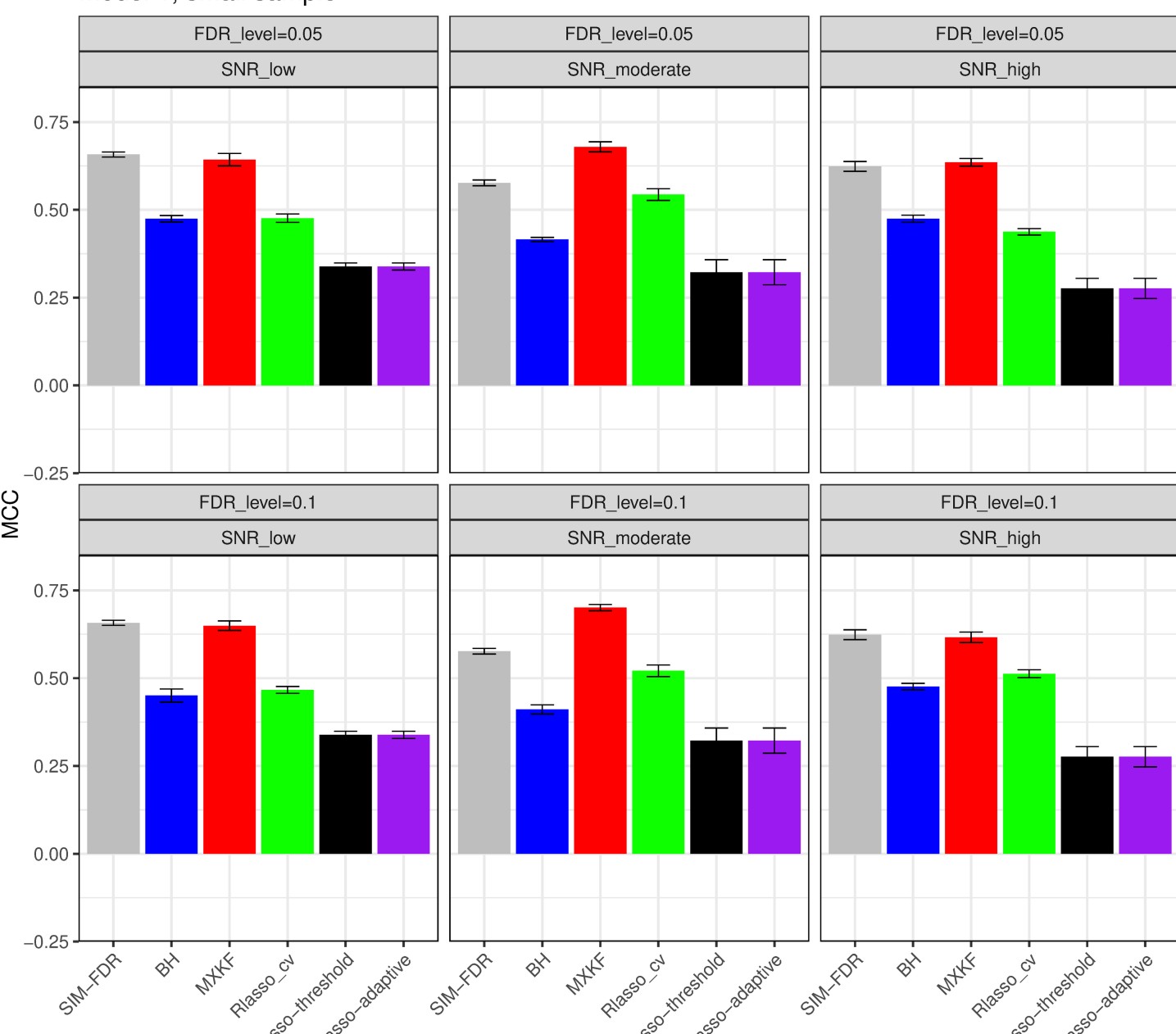

**Fig 16. Results for model 1 at small sample size case (n = 100)** MCCs are averaged over 100 replications, and their standard deviations (sd) are given on the top of the histogram.

which are all independently and randomly generated. We denoted this dataset as Tara-Noise, and our main goal is to identify the important taxa from the whole 2015 taxa (1015 real taxa and 1000 noise taxa). The noise taxa $Z_1$ and $Z_2$ can be viewed as false taxa which are not associated with the response. Given FDR level 0.1, the results of falsely selecting the noise taxa by all the methods are shown in Table 2. We observed that both BH and Rlasso-cv methods mistakenly selected a number of noise taxa while our method SIM-FDR did not select any noise

## Model 2, small sample

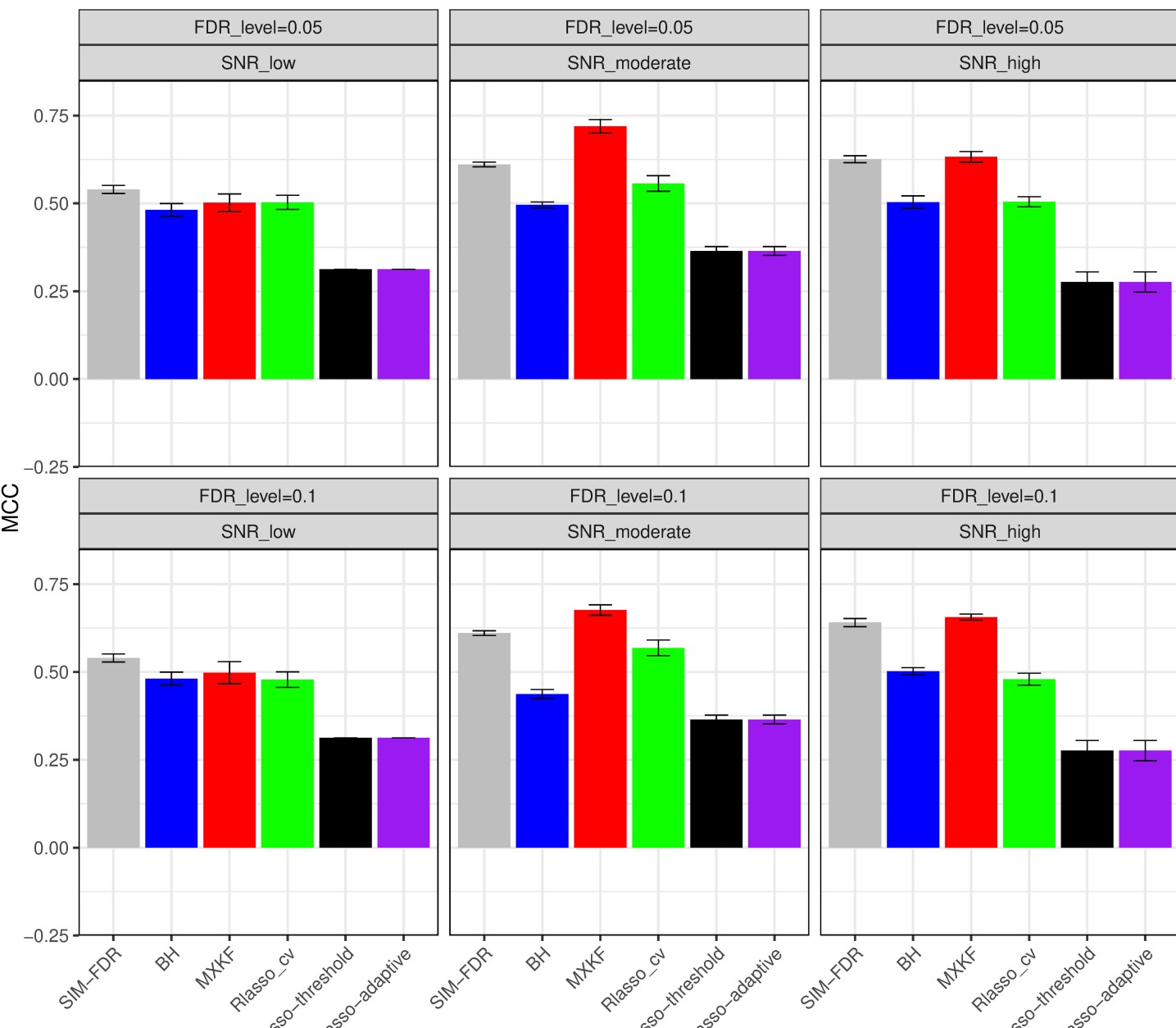

**Fig 17. Results for model 2 at small sample size case (n = 100)** MCCs are averaged over 100 replications, and their standard deviations (sd) are given on the top of the histogram.

taxa. This indicates that these two methods indeed produce more false discoveries, while our method achieves higher precision in detecting findings.

Given nominal FDR level 0.20, SIM-FDR method identified six taxa associated with the ocean salinity gradients. From Tables 3 and 4, the identified OTU197, OTU741 and OTU2043 by SIM-FDR come from the class *Alphaproteobacteria*, OTU1473 comes from the class *Deltaproteobacteria*, OTU1439 and OTU520 come from the class *Gammaproteobacteria*. For

## Model 3, small sample

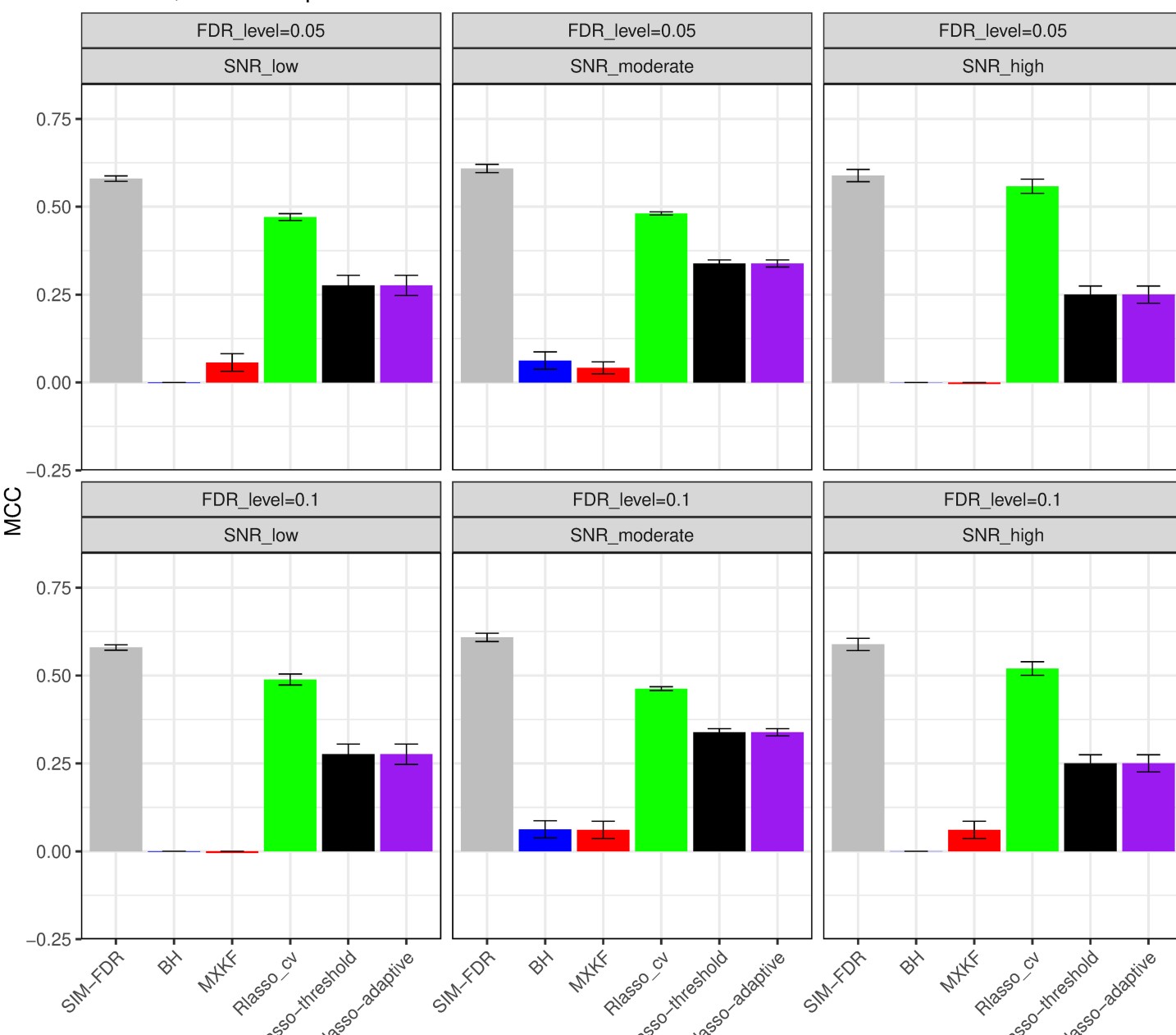

**Fig 18. Results for model 3 at small sample size case (n = 100)** MCCs are averaged over 100 replications, and their standard deviations (sd) are given on the top of the histogram.

the Tara data, Bien et al. [47] proposed a tree-aggregated predictive model and also used their method to conduct taxa selection. However, their selection result is very different from that of SIM-FDR. Thus, our results could offer some new perspective of the Tara data.

## Model 4, small sample

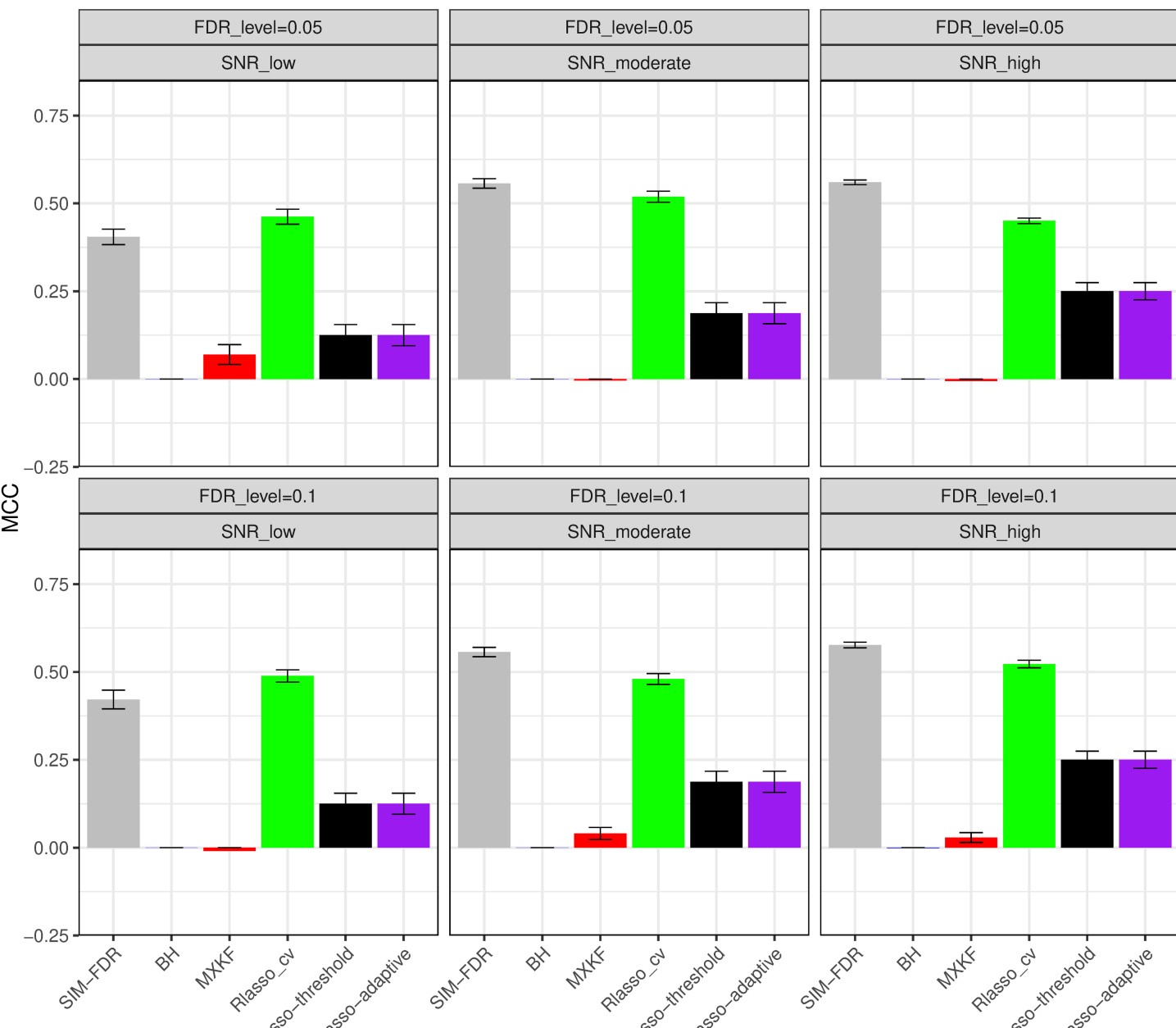

**Fig 19. Results for model 4 at small sample size case (n = 100)** MCCs are averaged over 100 replications, and their standard deviations (sd) are given on the top of the histogram.

### Head and neck squamous cell carcinoma data

Head and neck squamous cell carcinoma (HNSCC) is a prevalent and prognostically challenging cancer globally [48]. Since the release of the TCGA-HNSC dataset in 2015, over 1,000 related articles have been published. The original data includes a total of 18,409 gene expression values. A prescreening process was conducted using marginal Cox models, leading to the

## Model 5, small sample

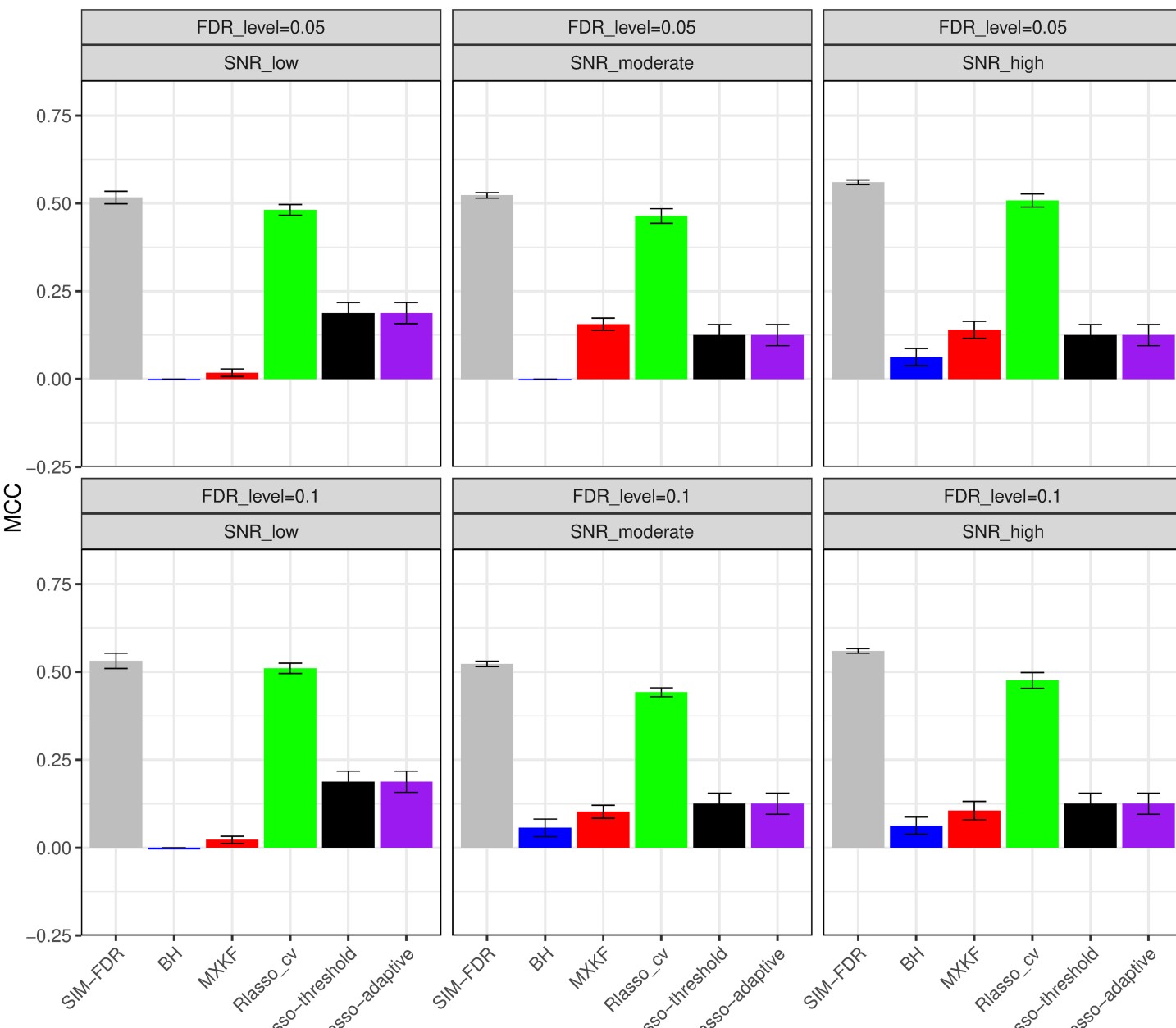

**Fig 20. Results for model 5 at small sample size case (n = 100)** MCCs are averaged over 100 replications, and their standard deviations (sd) are given on the top of the histogram.

selection of the top 2,000 genes with the smallest *p*-values for downstream analysis. The pre-processed HNSCC dataset, which contains 2,000 gene expression values, the logarithm of survival time, and a censoring indicator, can be downloaded from TCGA Provisional using the R packages *cgdsr* or *GEInter*. Additionally, the preprocessed HNSCC dataset can be available in S4 Dataset (Supporting information section).

## Model 5, moderate sample

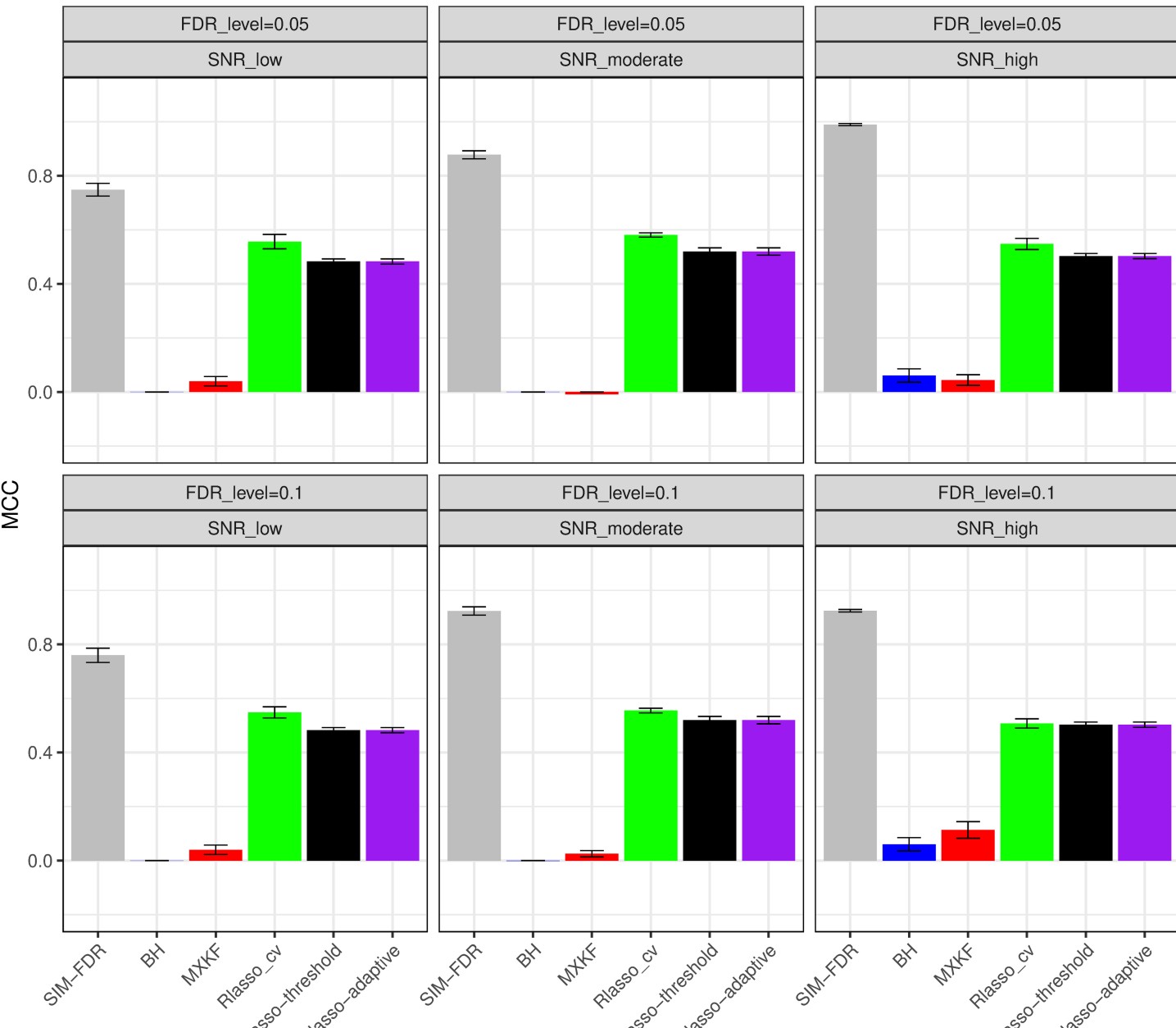

**Fig 21. Results for model 5 at moderate sample size case (n = 250)** MCCs are averaged over 100 replications, and their standard deviations (sd) are given on the top of the histogram.

Here, our objective was to identify potential genes associated with the survival time of head and neck squamous cell carcinoma (HNSCC). The results of this analysis are instrumental in understanding the molecular mechanisms underlying the occurrence and progression of HNSCC, and they hold significant implications for future treatment strategies. In the pre-processed HNSCC dataset, we collected 484 samples, with a censoring ratio of approximately 58%. This indicates that the true survival times for 58% of the samples remain unobserved

## Model 6, small sample

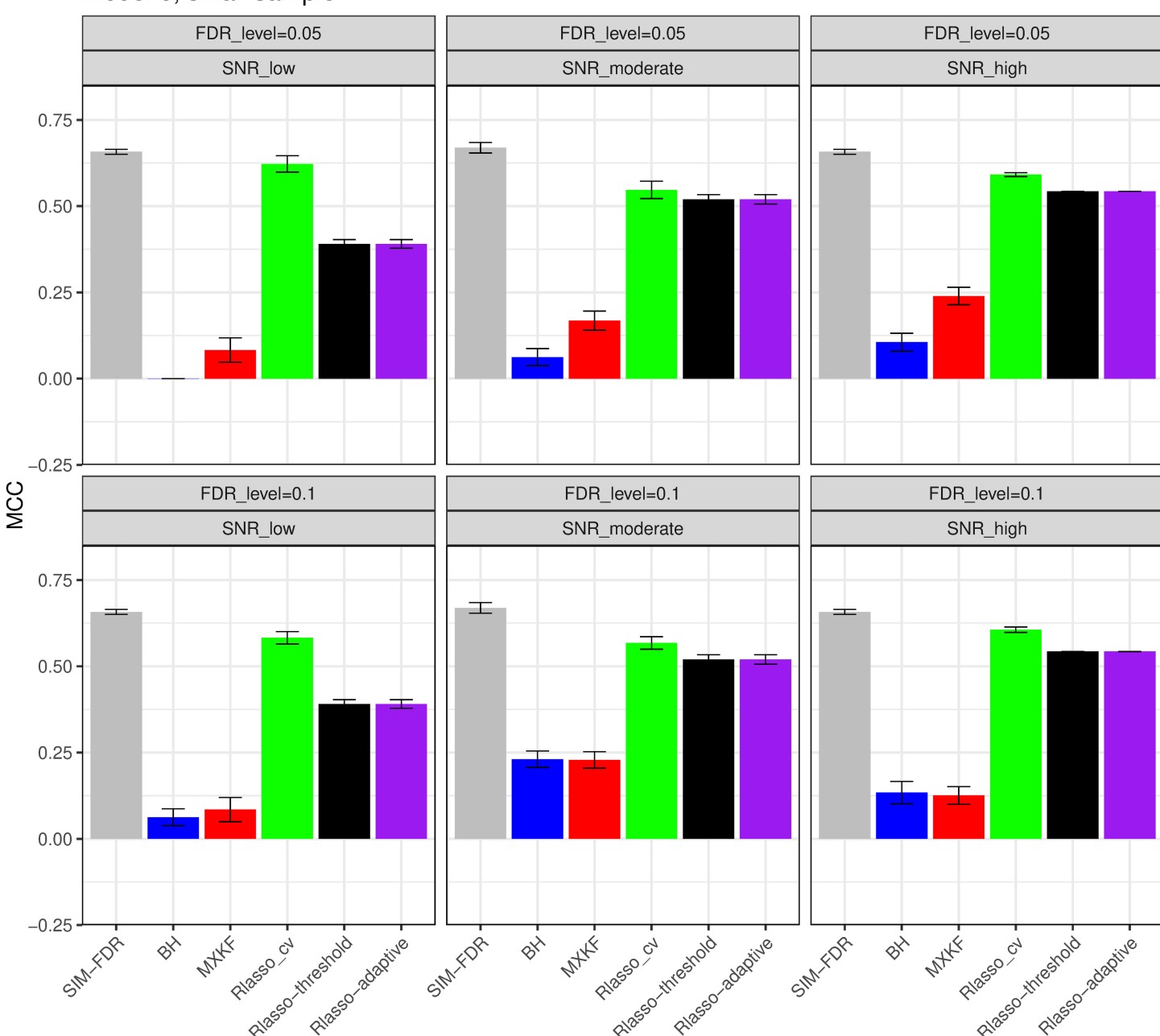

**Fig 22. Results for model 6 at moderate sample size case (n = 250)** MCCs are averaged over 100 replications, and their standard deviations (sd) are given on the top of the histogram.

and 42% of the samples are observed. To render this dataset suitable for our method, we utilized the observed survival time samples, resulting in 204 samples ($n = 484 \times 0.42 = 204$) for further analysis. We took 2000 genes as feature variables and employed the original survival time data, prior to logarithmic transformation, as the response variable. We then applied all relevant methods for gene selection on the dataset. Following the analysis of simulated data,

**Table 1. The number of selected taxa by all the methods under different nominal FDR levels.**

| FDR level | 0 | 0.02 | 0.04 | 0.07 | 0.09 | 0.11 | 0.13 | 0.16 | 0.18 | 0.20 |
|---|---|---|---|---|---|---|---|---|---|---|
| BH | 0 | 371 | 444 | 507 | 558 | 588 | 620 | 648 | 671 | 687 |
| MXKF | 0 | 0 | 0 | 0 | 0 | 0 | 0 | 0 | 0 | 0 |
| 01 SIM-FDR | 0 | 4 | 4 | 4 | 4 | 4 | 4 | 5 | 6 | 6 |
| Rlasso-cv | 0 | 68 | 68 | 68 | 68 | 68 | 68 | 68 | 68 | 68 |
| Rlasso-threshold | 0 | 2 | 2 | 2 | 2 | 2 | 2 | 2 | 2 | 2 |
| Rlasso-adaptive | 0 | 2 | 2 | 2 | 2 | 2 | 2 | 2 | 2 | 2 |

**Table 2. The number of selected noise taxa in 1000 noise taxa using the Tara-Noise data given the nominal FDR level ($\alpha = 0.1$).**

| Method | the number of selected noise taxa |
|---|---|
| BH | 30 |
| MXKF | 0 |
| SIM-FDR | 0 |
| Rlasso-cv | 2 |
| Rlasso-threshold | 0 |
| Rlasso-adaptive | 0 |

**Table 3. The information of six selected taxa by SIM-FDR at the FDR level 0.20.**

| Taxa Name | Rank | Kingdom | Phylum | Class | Order | Family |
|---|---|---|---|---|---|---|
| OTU197 | Life | Bacteria | Proteobacteria | Alphaproteobacteria | SAR11 clade | Surface |
| OTU741 | Life | Bacteria | Proteobacteria | Alphaproteobacteria | SAR11 clade | Surface |
| OTU2043 | Life | Bacteria | Proteobacteria | Alphaproteobacteria | Rickettsiales | S25-593 |
| OTU1473 | Life | Bacteria | Proteobacteria | Deltaproteobacteria | Desulfuromonadales | GR-WP33-58 |
| OTU1439 | Life | Bacteria | Proteobacteria | Gammaproteobacteria | KI89A clade | f__134 |
| OTU520 | Life | Bacteria | Proteobacteria | Gammaproteobacteria | E01-9C-26 marine group | f__69 |

**Table 4. The information of six selected taxa by SIM-FDR at the FDR level 0.20.**

| Taxa Name | Genus | Species |
|---|---|---|
| OTU197 | g__118 | AY664083.1.1206 |
| OTU741 | g__409 | EU801445.1.1438 |
| OTU2043 | g__681 | JN166192.1.1464 |
| OTU1473 | g__608 | EF574438.1.1503 |
| OTU1439 | g__601 | FR683972.1.1501 |
| OTU520 | g__304 | JF747664.1.1516 |

we set the same setting $\gamma = 2/3$ for SIM-FDR while let $B$ be 30 to stabilize the analysis results of SIM-FDR. We varied nominal FDR levels from 0 to 0.20 for this real data analysis.

The results are presented in Table 5. It is observed that the number of genes identified by the BH and Rlasso-cv methods exceeds that identified by SIM-FDR, while the MXKF, Rlasso-threshold, and Rlasso-adaptive methods did not identify any genes across all FDR levels. This may align with the simulation results from various scenarios, such as model 1 or model 2 under moderate sample conditions (Figs 1 and 2), which involve a sample size of ($n = 204$) and a gene count of ($p = 2000$). Figs 1 and 2 illustrate that the BH and Rlasso-cv methods exhibit higher actual FDRs and fail to control the FDR at the specified nominal levels. Therefore, the results in Table 5 may indicate that the BH and Rlasso-cv methods yield a greater number of false discoveries, whereas the SIM-FDR method may provide fewer but more accurate gene selection results.

**Table 5. The number of selected genes by all the methods under different nominal FDR levels.**

| FDR level | 0 | 0.02 | 0.04 | 0.07 | 0.09 | 0.11 | 0.13 | 0.16 | 0.18 | 0.20 |
|---|---|---|---|---|---|---|---|---|---|---|
| BH | 0 | 265 | 308 | 360 | 399 | 427 | 463 | 500 | 540 | 592 |
| MXKF | 0 | 0 | 0 | 0 | 0 | 0 | 0 | 0 | 0 | 0 |
| SIM-FDR | 0 | 3 | 3 | 3 | 3 | 3 | 7 | 7 | 7 | 7 |
| Rlasso-cv | 0 | 51 | 51 | 51 | 51 | 51 | 51 | 51 | 51 | 51 |
| Rlasso-threshold | 0 | 0 | 0 | 0 | 0 | 0 | 0 | 0 | 0 | 0 |
| Rlasso-adaptive | 0 | 0 | 0 | 0 | 0 | 0 | 0 | 0 | 0 | 0 |

Next, we introduced some simulated feature variables as non-association genes to further investigate the feature selection performance of all the methods. In details, 500 noise genes $Z_1$ with 204 samples are from the normal distribution $N(-1,2)$ and another 500 noise genes $Z_2$ with 204 samples are from the normal distribution $N(1,2)$, which are all independently and randomly generated. The genes $Z_1$ and $Z_2$ can be viewed as noise genes which are not associated with the survival time response. We denoted this dataset as HNSCC-Noise, and our main goal is to identify the important genes from the whole 3000 genes. Then we applied all the methods to the HNSCC-Noise dataset. The discovered genes are listed in Table 6. We observed that BH and Rlasso-cv mistakenly selected a number of noise genes while our method SIM-FDR did not select any noise genes. This indicates that these two methods indeed produce more false discoveries, while our method achieves higher precision in detecting findings.

Finally, at a significance level of $\alpha = 0.2$, Table 7 presents seven genes—SNX14, BICRAL, KIR3DL1, GRAPL, SEMA4D, IL20RA, and POLDIP2—that were detected using SIM-FDR. Notably, none of these genes have been reported in existing literature. Our findings may provide new insights into the HNSCC data, as these genes could enhance our understanding of the carcinogenic mechanisms associated with the cell cycle in HNSCC and may serve as potential biomarkers for the survival and treatment of HNSCC.

## Discussion and conclusion

In this paper, we first employ a more general single index model to fit omics data. This model is robust and can account for nonlinear associations with unknown distributional random errors and features. Next, based on this model, we develop an effective FDR control procedure for feature selection in high-dimensional single index models. We further apply this

**Table 6. The number of selected genes in 1000 noise genes using the HNSCC-Noise data given the nominal FDR level ($\alpha = 0.1$).**

| Method | the number of selected noise taxa |
|---|---|
| BH | 19 |
| MXKF | 0 |
| SIM-FDR | 0 |
| Rlasso-cv | 2 |
| Rlasso-threshold | 0 |
| Rlasso-adaptive | 0 |

**Table 7. The names of selected genes by SIM-FDR at the FDR level 0.2.**

| Genes' Names |
|---|
| SNX14, BICRAL, KIR3DL1, GRAPL, SEMA4D, IL20RA, POLDIP2 |

procedure to fine-map omics features while controlling the false discovery rate of selection. The results from simulated data indicate that when the linear or nonlinear model has heavy-tailed distributional random errors in moderate sample cases, the proposed SIM-FDR method significantly outperforms competing methods in power performance across nearly all scenarios while effectively controlling the FDR. In small sample cases, our method maintains actual FDRs at the nominal FDR levels for all scenarios, whereas nearly all competing methods fail to control the FDR. Compared to the SIM-FDR method, other methods may either be underpowered or yield inappropriate results, resulting in an inflated FDR relative to the nominal FDR threshold. Overall, these findings suggest that SIM-FDR demonstrates robust performance.

However, there are some aspects of our method that need to be discussed as follows.

1. Our approach primarily focuses on feature selection and does not involve estimating the link function or true coefficients of the original model for predicting the response. Once features are selected using our method, existing single index models for prediction can be utilized to fit the selected features, and the fitted model can subsequently be employed for prediction.

2. Estimating the unknown link function in high-dimensional single index models presents a significant challenge. Although our approach does not require estimating the link function, it assumes that the unknown link function is monotonic.

3. In the theoretical section, our method assumes that predictors are absolutely continuous and sub-gaussian. However, Rejchel et al. [32] have demonstrated through simulation studies that the rank-based approach can also be applied in scenarios with discrete-type predictors. This suggests that our method can be employed to analyze data with discrete-type predictors.

4. Specifically, our method is designed for single-index models and not for double-index or multi-index models. After careful consideration, we believe that combining sufficiency dimension reduction with the rank-based approach may address this issue. We hope to extend our method to accommodate double-index or multi-index scenarios in future work.

5. Our method is not suitable for handling binary-response and other discrete-type response datasets. We plan to explore ways to extend SIM-FDR for analyzing binary-response and other discrete-type response cases in future research.

Besides the above discussions, there has been considerable research interest in utilizing additional information (e.g., phylogenetic information) from microbiome data to enhance detection power while maintaining FDR control [49,50]. In the future, it will be of interest to incorporate such information into the SIM-FDR framework to further improve the detection power of controlled feature selection.

## Supporting information

**S1 File. The proofs of all the Theorems in Theory Properties section can be available.**
(PDF)

**S2 Dataset. The collected raw Tara data set.**
(ZIP)

**S3 Dataset. The preprocessed Tara data set.**
(ZIP)

**S4 Dataset. The preprocessed HNSCC dataset, which contains 2,000 gene expression values, the logarithm of survival time, and a censoring indicator, can also be available.** (ZIP)

## Acknowledgments

We thank all the women and men who agreed to participate in this study.

## Author contributions

**Data curation:** Yu-Fan Gao, Jian Xiao.

**Formal analysis:** Jian Xiao.

**Methodology:** Zhibo Chen, Zi-Tong Lu, Jian Xiao.

**Software:** Zi-Tong Lu, Jian Xiao.

**Writing – original draft:** Xue-Ting Song, Jian Xiao.

**Writing – review & editing:** Zhibo Chen, Xue-Ting Song, Jian Xiao.

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
