## [Decision Letter · Decision Letter 0]

14 May 2024

PONE-D-24-08088A novel model-free feature selection method with FDR control for omics-wide association analysisPLOS ONE

Dear Dr. Xiao,

Thank you for submitting your manuscript to PLOS ONE. After careful consideration, we feel that it has merit but does not fully meet PLOS ONE’s publication criteria as it currently stands. Therefore, we invite you to submit a revised version of the manuscript that addresses the points raised during the review process.

We look forward to receiving your revised manuscript.

Kind regards,

Mihye Ahn, Ph.D

Academic Editor

PLOS ONE

Journal Requirements:

"This work was supported by the National Natural Science Foundation of China (NSFC) (no.11801571, no.12171483 and no.61773401)."

Reviewers' comments:

Reviewer's Responses to Questions

**Comments to the Author**

1. Is the manuscript technically sound, and do the data support the conclusions?

Reviewer #1: Yes

Reviewer #2: Yes

2. Has the statistical analysis been performed appropriately and rigorously? 

Reviewer #1: Yes

Reviewer #2: Yes

3. Have the authors made all data underlying the findings in their manuscript fully available?

Reviewer #1: Yes

Reviewer #2: Yes

4. Is the manuscript presented in an intelligible fashion and written in standard English?

Reviewer #1: Yes

Reviewer #2: Yes

5. Review Comments to the Author

Reviewer #1: The authors have done a concrete representation of their work with clear explanation on their method with simulation results. The authors may consider comparing their method to more comparators, but the display they have looks OK. The English in the manuscript is mostly straightforward with some minor revisions needed in some sentences. The detailed comments are provided in the attached pdf file.

Reviewer #2: Here are some suggestions I’d like to provide on your revisions:

1. Please elaborate on the literature review in the introduction part. It’s crucial to see what previous research works have been done in this area with their cons and pros. It is also important to highlight the research gap your research addresses and to clarify the motivation behind your research. This will provide readers with a more solid understanding of novelty and importance of your research.

2. In the section discussing simulation study, I saw you compared with two other existing methods. It might be helpful to provide a more detailed description on them, such as the one developed by Benjamini and Hochberg. Explain why you want to perform comparisons with those two methods, how your chosen methods differ and what advantages/disadvantages they might have.

3. On Line 111, you reference an 'eta' in equation (2), but it does not appear to be present. Please either introduce 'eta' in the equation or revise the text to reflect the equation correctly.

4. Make sure to be consistent in notations throughout the manuscript. Specifically, the use of subscripts such as 'i' and 'j' in your method.

5. On Line 267, the 'Z' is mentioned. Please provide a clear explanation of 'Z' so that readers can understand its meaning.

6. PLOS authors have the option to publish the peer review history of their article (what does this mean?). If published, this will include your full peer review and any attached files.

Reviewer #1: **Yes: **Sungtaek Son

Reviewer #2: No

---

## [Decision Letter · Decision Letter 1]

20 Oct 2024

PONE-D-24-08088R1A novel robust  feature selection method with FDR control for omics-wide association analysisPLOS ONE

Dear Dr. Xiao,

Thank you for submitting your manuscript to PLOS ONE. After careful consideration, we feel that it has merit but does not fully meet PLOS ONE’s publication criteria as it currently stands. Therefore, we invite you to submit a revised version of the manuscript that addresses the points raised during the review process.

We look forward to receiving your revised manuscript.

Kind regards,

Mihye Ahn, Ph.D

Academic Editor

PLOS ONE

Journal Requirements:

Reviewers' comments:

Reviewer's Responses to Questions

**Comments to the Author**

1. If the authors have adequately addressed your comments raised in a previous round of review and you feel that this manuscript is now acceptable for publication, you may indicate that here to bypass the “Comments to the Author” section, enter your conflict of interest statement in the “Confidential to Editor” section, and submit your "Accept" recommendation.

Reviewer #1: All comments have been addressed

Reviewer #2: (No Response)

2. Is the manuscript technically sound, and do the data support the conclusions?

Reviewer #1: Yes

Reviewer #2: Yes

3. Has the statistical analysis been performed appropriately and rigorously? 

Reviewer #1: Yes

Reviewer #2: Yes

4. Have the authors made all data underlying the findings in their manuscript fully available?

Reviewer #1: Yes

Reviewer #2: Yes

5. Is the manuscript presented in an intelligible fashion and written in standard English?

Reviewer #1: Yes

Reviewer #2: Yes

6. Review Comments to the Author

Reviewer #1: The revised version of the paper looks much better than the initial version. I appreciate that the authors have accepted and reflected the recommendations. However, some more revision is needed due to some inconsistency and confusion in the use of the notations and expressions. Please see the comments in the attached file.

Reviewer #2: After the first revision, the previous comments have been partially addressed, and the content has become more comprehensive . However, there are still minor grammatical issues, and the use of professional language can be further improved. I recommend thoroughly proofreading the paper and revising it to make sure the use of precise academic language. Additionally, please double check all mathematical notations for correct usage, and remove any interpretation of notations that have not appeared in the context.

7. PLOS authors have the option to publish the peer review history of their article (what does this mean?). If published, this will include your full peer review and any attached files.

Reviewer #1: **Yes: **Sungtaek Son

Reviewer #2: No

---

## [Author Response · Author response to Decision Letter 2]

3 Dec 2024

Response to the comments of reviewer

Thank you very much for your helpful and constructive comments on our manuscript. We have amended our manuscript following your valuable suggestions. Specific replies to your comments are given below:

Line 146: "distributional" "distribution of".

Line 179: "I" "we".

Line 185: "gotten" "obtained".

Line 191: "usually do not satisfy asymptotically symmetric around zero property" "are usually not asymptotically symmetric around zero".

Line 224 and 228: I would suggest the authors distinguish the notations for the candidate features for samples and . For example, instead of using �the authors can consider and .

Response: Thanks very much! We have revised all the above problems in the new manuscript in red.

Line 237 - 239 "It is well known -- complex distribution variable": I would suggest adding references that discuss that bootstrap re-sampling can be used to obtain consistent estimate of the variance.

Response: Thanks very much! We have added the following previous references in the new manuscript in red.

1、DebashisKushary. Bootstrap Methods and Their Application. Technometrics, 2000, 42(2):216-217.

2、Efron B. Bootstrap Methods: Another Look at the Jackknife, Annals of Statistics, 1979, 7(1):1-26.

3、Johnson R W. An Introduction to the Bootstrap. Teaching Statistics, 2010, 23(2):49-54.

Line 299: "Theory properties" "Theoretical properties".

Line 302: Is be a vector in . Please specify. If so, the authors would need to use the transpose of , i.e., instead of .

Response: Thanks very much! We have revised all the above problems in the new manuscript in red.

Line 312-313: and would merely be observations, not "sets". Please use an alternative notation to denote sets.

Response: Thanks very much! We have revised all the above problems in the new manuscript in red.

Line 305 - 306: For real numbers, I suggest using . Also, please make the notations consistent. On line 306, there are two expressions for expectation: E and E . Also, is a real number or a function whose range is in �

Line 356: "non matter" "no matter".

Response: Thanks very much! We have revised all the above problems in the new manuscript in red.

---

## [Decision Letter · Decision Letter 2]

20 Feb 2025

PONE-D-24-08088R2A novel robust  feature selection method with FDR control for omics-wide association analysisPLOS ONE

Dear Dr. Xiao,

Thank you for submitting your manuscript to PLOS ONE. After careful consideration, we feel that it has merit but does not fully meet PLOS ONE’s publication criteria as it currently stands. Therefore, we invite you to submit a revised version of the manuscript that addresses the points raised during the review process.

Reviewer 1 has suggested a few minor revisions that need to be addressed. I believe that once these adjustments are made, your manuscript will meet the necessary requirements for approval. 

We look forward to receiving your revised manuscript.

Kind regards,

Mihye Ahn, Ph.D

Academic Editor

PLOS ONE

Journal Requirements:

Reviewers' comments:

Reviewer's Responses to Questions

**Comments to the Author**

1. If the authors have adequately addressed your comments raised in a previous round of review and you feel that this manuscript is now acceptable for publication, you may indicate that here to bypass the “Comments to the Author” section, enter your conflict of interest statement in the “Confidential to Editor” section, and submit your "Accept" recommendation.

Reviewer #1: All comments have been addressed

Reviewer #2: All comments have been addressed

2. Is the manuscript technically sound, and do the data support the conclusions?

Reviewer #1: Yes

Reviewer #2: Yes

3. Has the statistical analysis been performed appropriately and rigorously? 

Reviewer #1: Yes

Reviewer #2: Yes

4. Have the authors made all data underlying the findings in their manuscript fully available?

Reviewer #1: Yes

Reviewer #2: Yes

5. Is the manuscript presented in an intelligible fashion and written in standard English?

Reviewer #1: Yes

Reviewer #2: Yes

6. Review Comments to the Author

Reviewer #1: Thank you for the opportunity to review the revised manuscript on the robust feature

selection method with FDR control for omics-wide association analysis.

I appreciate the efforts put in by the authors for revising the manuscript based on the

reviewer’s comments. The contents are much clearer, and the mathematical expressions

are easier to follow.

Upon reading the paper line by line, I have only had some minor comments. These are not

necessarily crucial in following the writing, but still stand out and could be disturbing for

future readers to some extent. Please find the line-by-line comments in the attached pdf file. In particular,

the comment regarding Line 133 should be addressed.

Overall, the manuscript now has enriched content and appears to suffice the standards

required for publication in PLOS One. I recommend that the paper be accepted for

publication.

Reviewer #2: (No Response)

7. PLOS authors have the option to publish the peer review history of their article (what does this mean?). If published, this will include your full peer review and any attached files.

Reviewer #1: **Yes: **Sungtaek Son

Reviewer #2: No

---

## [Author Response · Author response to Decision Letter 3]

21 Feb 2025

Thank you very much for your helpful and constructive comments on our manuscript. We have amended our manuscript following your valuable suggestions. Specific replies to your comments are given below:

Overall, the manuscript now has enriched content and appears to suffice the standards required for publication in PLOS One. I recommend that the paper be accepted for publication.

Abstract :

Line 4: "associated relat ionships" → "association" Line 10: "can be connected" → "to be connected"

Response: Thanks very much! We have revised all the above problems in the new manuscript in red.

The main body of the manuscript:

Line 11: (Erase "is") omics features is much larger → omics features much larger

Line 11: (Add "which") also poses new → which also poses new

Line 17: (Erase "done by") can be done by based on → can be based on

Response: Thanks very much! We have revised all the above problems in the new manuscript in red.

Line 95: (Recommend using an alternative term for "notable", e.g., useful) hence it enjoys many notable theoretical properties. Thence it enjoys many useful theoretical properties

Response: Thanks very much! We have revised this problem in the new manuscript in red.

Line 101: to valid → to validate

Line 131: Add citation number for "Rejchel et al."

Response: Thanks very much! Sorry for my careless! We have revised this problem in the new manuscript in red.

Line 133: is not used in any expression above.

Response: Thanks very much! Sorry for my careless! We have deleted it.

Line 137: The subscript in the definition for should be instead of RP.

Line 304: For consistency, replace E() by E().

Response: Thanks very much! We have revised this problem in the new manuscript in red.

Line 342: asymptotically symmetry → asymptotically symmetric

Line 344: In Theorem 1, considering that is a vector,

should be revised. For example, it could be .

Response: Thanks very much! We have revised this problem in the new manuscript in red.

Line 395, 401, and 559: "following the work[]" can be elaborated with author names, for example, "following the work of Rejchel et al. [32]".

Line 556: " then" → "thus"?

Response: Thanks very much! We have revised this problem in the new manuscript in red

---

## [Decision Letter · Decision Letter 3]

4 May 2025

PONE-D-24-08088R3A novel robust  feature selection method with FDR control for omics-wide association analysisPLOS ONE

Dear Dr. Xiao,

Thank you for submitting your manuscript to PLOS ONE. After careful consideration, we feel that it has merit but does not fully meet PLOS ONE’s publication criteria as it currently stands. Therefore, we invite you to submit a revised version of the manuscript that addresses the points raised during the review process.

We look forward to receiving your revised manuscript.

Kind regards,

Wan-Tien Chiang

Academic Editor

PLOS ONE

Journal Requirements:

Reviewers' comments:

Reviewer's Responses to Questions

**Comments to the Author**

1. If the authors have adequately addressed your comments raised in a previous round of review and you feel that this manuscript is now acceptable for publication, you may indicate that here to bypass the “Comments to the Author” section, enter your conflict of interest statement in the “Confidential to Editor” section, and submit your "Accept" recommendation.

Reviewer #1: All comments have been addressed

Reviewer #3: All comments have been addressed

2. Is the manuscript technically sound, and do the data support the conclusions?

Reviewer #1: Yes

Reviewer #3: Yes

3. Has the statistical analysis been performed appropriately and rigorously? 

Reviewer #1: Yes

Reviewer #3: Yes

4. Have the authors made all data underlying the findings in their manuscript fully available?

Reviewer #1: Yes

Reviewer #3: Yes

5. Is the manuscript presented in an intelligible fashion and written in standard English?

Reviewer #1: Yes

Reviewer #3: Yes

6. Review Comments to the Author

Reviewer #1: Thank you for the opportunity to read and review your work. I carefully read the revised manuscript and I do not have any further comments.

Reviewer #3: 

1. Introduction part: the authors utilize rank-based approach and SDA approach to develop a robust feature selection method for a single index model (SIM). Meanwhile, the authors introduce several methods of FDR controlled feature selection: Knockoff filter-based approach, Symmetrized data aggregation (SDA) approach, Stability selection approach. Based on these conceptions, can authors provide rationelle for integrate rank/SDA methods into SIM in the introduction part?

2. Results part: as observed from results, SIM-FDR is not always ranked to be highest in both aspects of FDR as well as Power. Can authors use a kind of overall/balanced evaluation method to judge that SIM-FDR is best when compared to other methods?

3. Results part: since massive human genomic data sets makes the dimensionality of omics features much larger than its sample size, can authors

also use real human genomic data sets to test performance of SIM-FDR?

4. Discussion: list ponits as above

7. PLOS authors have the option to publish the peer review history of their article (what does this mean?). If published, this will include your full peer review and any attached files.

Reviewer #1: **Yes: **Sungtaek Son

Reviewer #3: No

---

## [Author Response · Author response to Decision Letter 4]

19 Jun 2025

Response to the comments of reviewer

Thank you very much for your helpful and constructive comments on our manuscript. We have amended our manuscript following your valuable suggestions. Specific replies to your comments are given below:

Reviewer #1: Thank you for the opportunity to read and review your work. I carefully read the revised manuscript and I do not have any further comments.

Reviewer #3:

1. Introduction part: the authors utilize rank-based approach and SDA approach to develop a robust feature selection method for a single index model (SIM). Meanwhile, the authors introduce several methods of FDR controlled feature selection: Knockoff filter-based approach, Symmetrized data aggregation (SDA) approach, Stability selection approach. Based on these conceptions, can authors provide rationelle for integrate rank/SDA methods into SIM in the introduction part?

Response: Thanks very much! We have provided some discussions in the revised manuscript in red between the lines 94 and 119.

2. Results part: as observed from results, SIM-FDR is not always ranked to be highest in both aspects of FDR as well as Power. Can authors use a kind of overall/balanced evaluation method to judge that SIM-FDR is best when compared to other methods?

Response: Thanks very much!

1. We first introduced the new measurement: Matthews correlation coefficient (MCC) to evaluate the performance of feature selection. MCC measures the overall accuracy of selection for true positives (TP), false negatives (FN), true negatives (TN), and false positives (FP), with a larger value indicating overall better selection (Baldi et al. 2000; Chicco and Jurman 2020). We have added some descriptions about MCC between the lines 476 and 479 in the revised manuscript in red.

2. The simulation results evaluated via the Matthews correlation coefficient (MCC) are shown in the revised manuscript in red between the lines 560 and 576.

3. Results part: since massive human genomic data sets makes the dimensionality of omics features much larger than its sample size, can authors also use real human genomic data sets to test performance of SIM-FDR?

Response: Thanks very much! We utilized the real Head and neck squamous cell carcinoma (HNSCC) data to further evaluate the performance of the proposed method. The Cancer Genome Atlas profiled 279 head and neck squamous cell carcinomas (HNSCC) to provide a comprehensive landscape of somatic genomic alterations. We have added the analysis of HNSCC data into the revised manuscript in red.between the lines 641 and 693.

4. Discussion: list ponits as above

Response: Thanks very much! We have reconstructed the “Discussion” section in the revised manuscript in red between the lines 694 and 737.

---

## [Decision Letter · Decision Letter 4]

29 Jul 2025

A novel and robust  feature selection method with FDR control for omics-wide association analysis

PONE-D-24-08088R4

Dear Dr. Xiao,

We’re pleased to inform you that your manuscript has been judged scientifically suitable for publication and will be formally accepted for publication once it meets all outstanding technical requirements.

Kind regards,

Austin W.T. Chiang

Academic Editor

PLOS ONE

Additional Editor Comments (optional):

Reviewers' comments:

Reviewer's Responses to Questions

**Comments to the Author**

1. If the authors have adequately addressed your comments raised in a previous round of review and you feel that this manuscript is now acceptable for publication, you may indicate that here to bypass the “Comments to the Author” section, enter your conflict of interest statement in the “Confidential to Editor” section, and submit your "Accept" recommendation.

Reviewer #3: All comments have been addressed

2. Is the manuscript technically sound, and do the data support the conclusions?

Reviewer #3: Yes

3. Has the statistical analysis been performed appropriately and rigorously? 

Reviewer #3: Yes

4. Have the authors made all data underlying the findings in their manuscript fully available?

Reviewer #3: Yes

5. Is the manuscript presented in an intelligible fashion and written in standard English?

Reviewer #3: Yes

6. Review Comments to the Author

Reviewer #3: All questions have been well answered.

1. Rationelle for SIM-FDR have been discussed;

2. MCC for SIM-FDR evaluation is suitable;

3. HNSCC data usage is suitable;

4. All points were listed.

We have learnt lots! Thank you!

7. PLOS authors have the option to publish the peer review history of their article (what does this mean?). If published, this will include your full peer review and any attached files.

Reviewer #3: No

---

## [Editor Report · Acceptance letter]

PONE-D-24-08088R4

PLOS ONE

Dear Dr. Xiao,

I'm pleased to inform you that your manuscript has been deemed suitable for publication in PLOS ONE. Congratulations! Your manuscript is now being handed over to our production team.

Kind regards,

on behalf of

Dr. Wan-Tien Chiang

Academic Editor

PLOS ONE